# MTU-Bench: A Multi-Granularity Tool-Use Benchmark for Large Language Models

**Pei Wang**[1*], **Yanan Wu**[1*], **Zekun Wang**[1*], **Jiaheng Liu**[1,2†],
**Xiaoshuai Song**[1], **Zhongyuan Peng**[1,3], **Ken Deng**[1], **Chenchen Zhang**[1], **Jiakai Wang**[1],
**Junran Peng**[3], **Ge Zhang**[4], **Hangyu Guo**[1], **Zhaoxiang Zhang**[3], **Wenbo Su**[1], **Bo Zheng**[1]
[1]Alibaba Group, [2]Nanjing University,
[3]University of Chinese Academy of Sciences, [4]University of Waterloo
{yupei.wp, lixing.wyn, ljh411989}@alibaba-inc.com

## ABSTRACT

Large Language Models (LLMs) have displayed massive improvements in reasoning and decision-making skills and can hold natural conversations with users. Recently, many tool-use benchmark datasets have been proposed. However, existing datasets have the following limitations: (1). Insufficient evaluation scenarios (e.g., only cover limited tool-use scenes). (2). Extensive evaluation costs (e.g., GPT API costs). To address these limitations, in this work, we propose a multi-granularity tool-use benchmark for large language models called MTU-Bench. For the "multi-granularity" property, our MTU-Bench covers five tool usage scenes (i.e., single-turn and single-tool, single-turn and multiple-tool, multiple-turn and single-tool, multiple-turn and multiple-tool, and out-of-distribution tasks). Besides, all evaluation metrics of our MTU-Bench are based on the prediction results and the ground truth without using any GPT or human evaluation metrics. Moreover, our MTU-Bench is collected by transforming existing high-quality datasets to simulate real-world tool usage scenarios, and we also propose an instruction dataset called MTU-Instruct data to enhance the tool-use abilities of existing LLMs. Comprehensive experimental results demonstrate the effectiveness of our MTU-Bench. Code and data will be released at https://github.com/MTU-Bench-Team/MTU-Bench.git.

## 1 INTRODUCTION

Since the release of large language models (LLMs) such as GPT-4 (OpenAI, 2023), Natural Language Processing (NLP) has entered a new wave of advancements, even being considered as the spark of Artificial General Intelligence (AGI) (Bubeck et al., 2023). Recently, there has been a surge of research focused on enabling LLMs to interface with external tools, such as calculators (Cobbe et al., 2021), search engines (Schick et al., 2023), and booking service APIs (Qin et al., 2023b). This approach, referred to as Tool Learning (Schick et al., 2023; Qin et al., 2023a;b; Wang et al., 2023), allows LLMs to not only accurately perform precise calculations, but also maintain up-to-date information. Furthermore, it enables LLMs to function as end-to-end AI assistants that are capable of fulfilling real-world user needs such as booking hotels or ordering food. Thus, Tool Learning is a critical step to transform LLMs into general AI agents.

Previous works have explored to stimulate the ability to call tools for LLMs (Schick et al., 2023; Qin et al., 2023b; Zhuang et al., 2023; Tang et al., 2023; Paranjape et al., 2023; Li et al.,

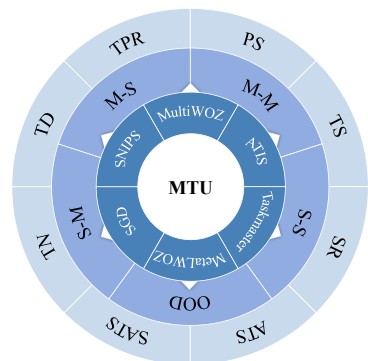

Figure 1: The circles from inside to outside represent the data source, scenes included in **MTU-Bench**, and the automatic evaluation metrics in **MTU-Eval**.

---

*First three authors contributed equally. [†] Corresponding Author: Jiaheng Liu.

Table 1: Comparison of various tool-use benchmark datasets. "Auto. Eval." denotes "automatic evaluation without GPT". "**S-S**", "**S-M**", "**M-S**", and "**M-M**" denote single-turn single-tool, single-turn multi-tool, multi-turn single-tool, and multi-turn multi-tool, respectively. "OOD" refers to whether the benchmark accounts for an out-of-distribution setting, where the test set consists of domains different from those in the training set. "Real-World" means whether the dialogues in the benchmark are sampled from real-world scenarios. The numbers in the evaluation range mean: ① tool selection, ② parameter selection, ③ dialogue-level success rate, ④ turn-based success rate, ⑤ tool number, and ⑥ tool order.

| Dataset | #Dialogues | #Tools | #Turn-#Tool | Real-World | Auto. Eval. | Eval. Range | Train | Test | OOD |
|---|---|---|---|---|---|---|---|---|---|
| **MetaTool** (Huang et al., 2024) | 21,127 | 199 | S-S, S-M | ✗ | ✓ | ①②③ | ✗ | ✓ | ✗ |
| **API-Bank** (Li et al., 2023) | 2,202 | 2,211 | S-S, S-M, M-S, M-M | ✗ | ✓ | ①②③ | ✓ | ✓ | ✗ |
| **ToolLLM** (Qin et al., 2023b) | 12,657 | 16,464 | S-S, S-M | ✗ | ✗ | ②③ | ✓ | ✓ | ✓ |
| **API-Bench** (Patil et al., 2023) | 17,002 | 1,645 | S-S | ✗ | ✓ | ②③ | ✓ | ✓ | ✗ |
| **ToolAlpaca** (Tang et al., 2023) | 3,938 | 400 | S-S, S-M, M-S, M-M | ✗ | ✗ | ①②③④ | ✓ | ✓ | ✗ |
| **ToolQA** (Zhuang et al., 2023) | 1,530 | 13 | S-S, S-M | ✗ | ✓ | ②③ | ✗ | ✓ | ✗ |
| **T-Eval** (Chen et al., 2024) | 23,305 | 15 | S-S, S-M | ✗ | ✓ | ①②③ | ✗ | ✓ | ✗ |
| **GTA** (Wang et al., 2024) | 229 | 14 | S-S, S-M | ✓ | ✓ | ①②③ | ✗ | ✓ | ✗ |
| **MTU-Bench** (Ours) | 54,798 | 136 | S-S, S-M, M-S, M-M | ✓ | ✓ | ①②③④⑤⑥ | ✓ | ✓ | ✓ |

2023). For example, (Schick et al., 2023) propose to convert tool calls into text spans, such as <API>SOME PARAMETER KEY-VALUE PAIRS</API> to denote the tool name and parameters with an additional special token (i.e., </API>) to show the initiation of a tool execution. Moreover, the recent works (i.e., ToolBench (Xu et al., 2023; Qin et al., 2023b), APIBench (Patil et al., 2023), and API-Bank (Li et al., 2023)) have investigated instruction tuning data or evaluation for tool-use.

However, we observe that exhibit several limitations to varying degrees as shown in Table 1: (1) some do not consider multi-turn dialogue scenarios (Patil et al., 2023; Xu et al., 2023; Zhuang et al., 2023); (2) some do not address multi-tool usage scenarios (Tang et al., 2023; Patil et al., 2023; Li et al., 2023; Xu et al., 2023); (3) several works use external API tools to deduce user instructions, but these synthesized instructions often do not accurately align with actual real-world user needs (Qin et al., 2023b); (4) many of them rely on GPT for the evaluation, leading to heavy evaluation costs (Qin et al., 2023b; Tang et al., 2023); and (5) many do not comprehensively assess fine-grained aspects of tool-use (Li et al., 2023; Qin et al., 2023b; Patil et al., 2023), such as the accuracy of tool call orders, complex tool calls involving inheritance relationships, per-dialogue turn accuracy of tool and parameter selection, etc.

To remedy these issues, in Figure 1, we introduce **MTU-Bench** (Multi-Granularity Tool-Use Benchmark), which comprises both **MTU-Instruct** for training and **MTU-Eval** for evaluation. As illustrated in Figure 2, we sample real-world user instructions from various existing open-source dialogue datasets such as MultiWOZ (Budzianowski et al., 2018) and SGD (Rastogi et al., 2020a; Lee et al., 2022). After instruction clustering, the detected user intents and slot filling are leveraged to synthesize API calls using GPT-4 (OpenAI, 2023). The synthesized data includes the thoughts, the actions (*i.e.*, tool names), the action parameters, and the observations (*i.e.*, the generated API execution results). This data forms our MTU-Bench dataset. Following meticulous quality verification by GPT-4 and manual check, we split the MTU-Bench data into training and testing splits, involving 54798 dialogues in total, as well as 136 tools. In our MTU-Eval, we propose a series of fine-grained metrics such as tool selection accuracy, parameter selection accuracy, success rate, turn success rate, task process rate, tool number accuracy, tool order accuracy, etc., to evaluate the tool-use abilities in a comprehensive manner, where the GPT API costs are not needed for evaluation. Moreover, we also

pick out a hard subset from the test split to include more complex tool-use scenarios such as easily confusable tools, nonsensical or noisy tools, tool parameter updating, etc.

Finally, by fine-tuning LLaMA-3 (Dubey et al., 2024) on MTU-Bench, we find that our resulting model, MTU-LLaMA, performs the best in various scenarios and metrics, demonstrating the effectiveness of our MTU-Instruct.

In summary, our contributions are as follows: (1). **MTU-Bench**: We introduce a novel automated data synthesis pipeline designed to derive high-quality, fine-grained tool-use datasets from pre-existing task-oriented dialogue datasets. This pipeline facilitates the creation of MTU-Bench, comprising MTU-Instruct for training purposes and MTU-Eval for evaluation. (2). **MTU-Instruct and MTU-Eval**: We introduce the high-quality and diverse instruction tuning dataset, MTU-Instruct, to improve models' tool-use capabilities in real-world scenarios. Additionally, we propose a novel automatic evaluation framework, MTU-Eval, which assesses various tool-use settings through comprehensive and fine-grained metrics, free of GPT-based evaluators. (3). **MTU-LLaMA and Experimental Findings**: After instruction tuning on MTU-Instruct, we obtain a strong open-source model for tool-use, MTU-LLaMA. Our comprehensive experiments reveal several findings regarding the tool-use capabilities of LLMs, particularly in terms of multi-turn dialogue scenarios, multi-tool settings, and error cases. These findings offer valuable insights for advancing tool-use in LLMs.

## 2 MTU-BENCH

The MTU-Bench involves both MTU-Instruct for training and MTU-Eval for evaluation. We first present the data construction and analysis in §2.1, and then show the evaluation procedure in §2.2.

### 2.1 MTU-INSTRUCT

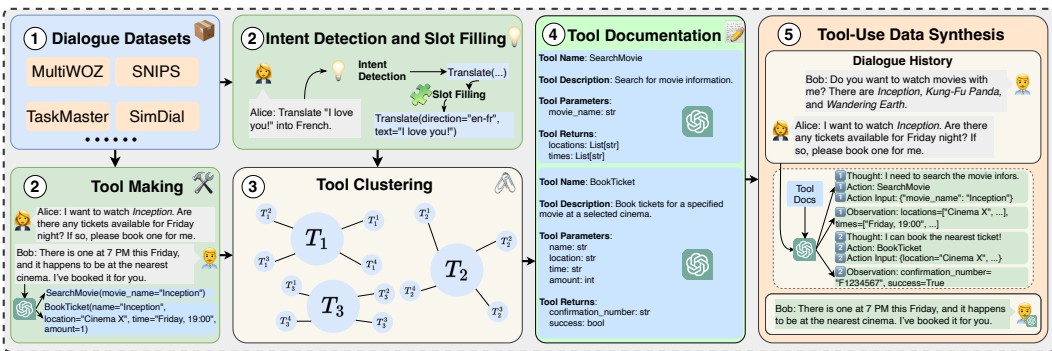

Figure 2: The workflow for MTU-Instruct construction. It involves five steps: (1) Data Collection, (2) Tool Creation, (3) Tool Clustering, (4) Tool Documentation, and (5) Tool-Use Synthesis.

#### 2.1.1 DATA CONSTRUCTION

Many previous methods based on scraped API documentation and GPT-4 inspired synthesis faced limitations due to a lack of data diversity, resulting in weak generalization capabilities. Inspired by the mapping relationships between intents and APIs, as well as between slots and tool parameters, we designed an automated data synthesis pipeline that transforms traditional dialogue datasets into tool-use datasets. To enhance data diversity, we collected datasets from multiple NLP datasets and standardized them into a unified tool documentation. The diversity is illustrated in Figure 3. As shown in Figure 2, the construction of MTU-Bench involves five primary steps: (1) collecting task-oriented dialogue datasets, particularly those containing intents and slots, (2) creating tools through grammar-based transformations or GPT-4 synthesis, (3) clustering the tools based on their similarities, (4) generating tool documentation using GPT-4, and (5) synthesizing tool-use samples consisting of thoughts, actions(tool calls), action inputs, observations, and adjusted responses based on the dialogue and tool documentation, followed by a holistic validation process.

**Data Collection.** To improve the diversity of our dataset, we collect several open-source task-oriented dialogue datasets as our data sources. These datasets focus on dialogues for specific tasks

such as flight reservations or movie bookings, which are highly suitable for synthesizing tool-use data. The multi-turn dialogue datasets include MultiWOZ (Budzianowski et al., 2018), SGD (Rastogi et al., 2020b), TaskMaster (Byrne et al., 2019) and MetaLWOZ (Shalyminov et al., 2020). The single-turn dialogue datasets include ATIS (Hemphill et al., 1990) and SNIPS (Siddhant et al., 2018). They provide diverse task-oriented dialogues across various domains, real-world conversation and fine-grained annotation, encompassing both single-turn and multi-turn dialogues.

**Tool Creation.** We employ two approaches to create tools. (1) **Grammar-based creation**. For dialogue datasets that already have detected intents and filled slots, we directly convert the intents into tool names and the slots into tool parameters. For example, in the user query *"find a flight from charlotte to las vegas"*, the intent *"Flight"* will convert to the tool name, and the slot *"from_location=charlotte, to_location=las vegas"* will convert to the parameters, resulting in the tool call Flight(from_location="charlotte",to_location="las vegas"). (2) **LLM-based creation**. For dialogue datasets without predefined intents or slots, we utilize GPT-4 to make the tools. Based on the contextual situation, we categorize it into five scenarios: information missing, information exists, information confirmed, aimless chatting and specific API call. Provided with the historical dialogue context and the current round of conversation, especially the assistant's response, LLM needs to determine which situation belongs to the current situation. **Information missing.** If the response is asking for important information, it should be the situation of missing information, no tool call should be made, and necessary parameters should be accumulated for related tools. **Information exists.** When the LLM can provide a response based solely on the information from the dialogue history, no tool call will be made and the model can directly reply. **Information confirmed.** When the assistant is confirming information (e.g.,*Would you like to confirm this flight reservation?*), this is classified as "information to be confirmed". **Aimless chatting.** If the scenario pertains to aimless chatting or situations that do not necessitate tool invocation, no tool call should be made. **Specific API call.** Only if the LLM determines that an API call is necessary to fulfill the user's request, it is encouraged to generate an appropriate pseudo-tool for invocation, along with a corresponding description and parameters of this tool.

**Tool Clustering.** Due to the diversity in both the intent detection and slot-filling strategies, as well as the creation of specific tools based on LLM, the synthesized tool set can be highly redundant. For example, tools like "search_movie" and "find_movie" may have different names but essentially perform the same function. To address this redundancy, we introduce a tool clustering phase. Specifically, we cluster the tool names based on InsTag(Lu et al., 2023) with a fixed distance threshold. Then, all tool names and their parameter names are standardized to the centroid of their respective cluster, resulting in a reduction ratio of 20:1.

**Tool Documentation.** To enable models to use specific tools effectively, we compile all tool usage into a comprehensive tool document. This document allows the model to determine the appropriate tool names and their usage. We prompt the GPT-4 model to write a description for each tool generated in the previous step, along with information about its parameters(required and optional) and returns. The collection of these tool entries forms the final tool document, which is included as part of the LLMs' context. Please refer to Appendix B for the details.

**Tool-Use Data Synthesis.** In this step, we convert all samples from task-oriented dialogue datasets into tool-use dataset with GPT-4, following the format of ReAct (Yao et al., 2023). We provide the dialogue history and the tool document generated in the previous steps as context for GPT-4, and then prompt it to generate three key components: (1) **thought**: the reasoning process behind the tool selection, (2) **action**: the name of the tool being invoked, and (3) **action input**: the parameters used in the tool call along with their values. This chain of thought prompting technique enhances the model's ability to reason over the most appropriate tool and accurately input the parameter values. We also allow the model to generate any additional parameters needed that are beyond those listed in the tool document, to ensure completeness and flexibility of tool-use.

We further ask the GPT-4 to simulate tool execution, generating **observation** (i.e., the results of the simulated tool execution) and then produce the final **response** for the current dialogue turn. The observations are aligned with the return information in the tool document and are generated in a structured format, such as a JSON dictionary. The model then formulates a response based on the observations, either to report the status of the tool execution or to complete the dialogue turn.

To ensure data quality, we apply various quality filters and adjustments, including heuristic rules, GPT-4, and manual annotation. For the training set, we use GPT-4 to verify the accuracy and necessity of tool selection, check parameter matching, adjust thoughts, rewrite response and ensure consistency in the tool definitions. Through GPT-4, we filter out about 10% of defective samples. For the test set, we hired multiple experts to conduct manual quality checks based on similar principles. Each sample was checked by three experts, and the differences in labeling were determined by the fourth expert.

Based on the number of dialogue turns and the number of tools in each dialogue, the synthesized data can be categorized into four types: (1) Single-turn Single-tool (**S-S**), (2) Single-turn Multi-tool (**S-M**), (3) Multi-turn Single-tool (**M-S**), and (4) Multi-turn Multi-tool (**M-M**).

For more detailed information about the construction of MTU-Bench, including prompt templates and the tool documentation, please refer to the Appendix B.

### 2.1.2 DATA ANALYSIS

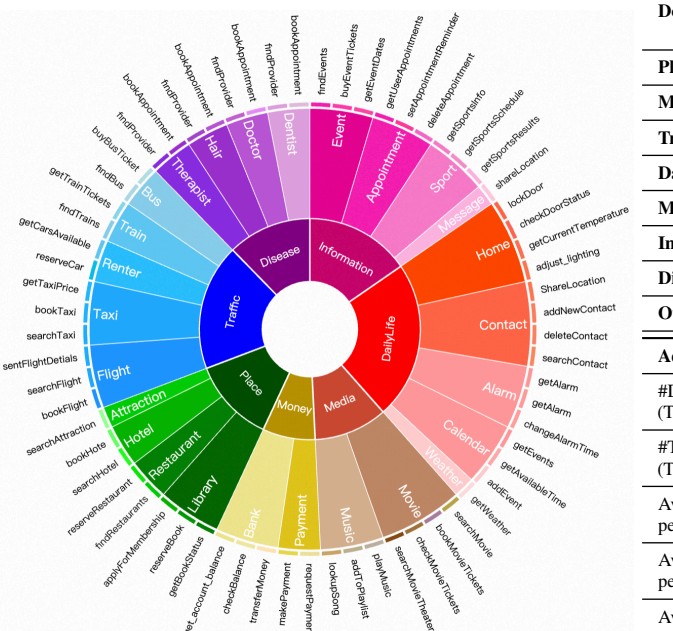

Figure 3: The domain distribution of tools.

| Domain | | #Topics | #Tools | #Dialogues (Train/Test) |
|---|---|---|---|---|
| **Place** | | 4 | 27 | 29,667 / 150 |
| **Media** | | 2 | 18 | 13,730 / 92 |
| **Traffic** | | 5 | 22 | 17,720 / 89 |
| **Daily Life** | | 5 | 31 | 23,706 / 163 |
| **Money** | | 2 | 10 | 1,740 / 32 |
| **Information** | | 4 | 12 | 9,125 / 75 |
| **Disease** | | 4 | 2 | 3,371 / 125 |
| **Others** | | 5 | 14 | - / 68 |
| **Additional Statistics** | | | | |
| #Dialogues (Train/Test) | | | 54,367 / 431 | |
| #Tools (Train/Test) | | | 122 / 14 | |
| Avg. Turns per Dialogue | | | 2.6 | |
| Avg. Tools per Dialogue | | | 5.6 | |
| Avg. Tools per Turn | | | 2.2 | |

Figure 4: Statistics of MTU-Bench.

**Statistics.** Figure 4 shows the statistical metrics. Our MTU-Bench is particularly designed to include multi-tool and multi-turn settings, as well as *real-world* domains and topics.

**Diversity.** Figure 3 illustrates the domain and topic distribution of the tools within our MTU-Bench. Figures 10 and 11 depict the distributions of dialogue turn counts, word counts, tool numbers, and other length-related metrics across both training and testing splits. These figures underscore the diversity of our MTU-Bench in terms of length and topic distribution.

We refer the readers to Appendix C for a more detailed data analysis.

### 2.2 MTU-EVAL

We propose **MTU-Eval**—the first evaluation framework that encompasses multiple levels of difficulty, diverse domains and cases of tool-use, varying numbers of dialogue turns and tools, as well as multifaceted evaluation by considering various granularities and aspects of LLM tool-use.

We present MTU-Eval in two parts: (1) Test Set Splitting and (2) Evaluation Metrics.

### 2.2.1 Test Set Splitting

We construct two distinct test sets from MTU-Bench through manual sampling: (1) the normal test set and (2) the more challenging hard test set.

**Normal/Hard Test Set.** MTU-Bench includes data from 31 different topics, such as weather-related and calendar-related tasks. Initially, we select data from 5 topics as an Out-of-Distribution (OOD) test split. From the remaining 26 topics, we further split the data into a training set and an in-domain test set. For more challenging evaluations, we manually curate a hard test set, which includes more complex tool-use cases, such as those involving long tool parameters, easily confusable tools, parameter value updating, scenarios with a large number of tools, etc., as listed in Appendix D.

### 2.2.2 Metrics

**For Single-Tool Scenarios (S-S & M-S).** For scenarios where there is just a single tool involved, we evaluate two metrics: **Tool Selection Accuracy (TS)** and **Parameter Selection Accuracy (PS)**.

**For Multi-Turn Scenarios (M-S & M-M).** In multi-turn dialogues, the following metrics are introduced: (1) **Success Rate (SR):** A binary metric where the entire dialogue is considered successful for tool-use: (=1) only if there are no errors throughout all turns; otherwise, it is considered unsuccessful (=0). (2) **Averaged Turn Success Rate (ATS):** We first evaluate each dialogue turn with the tool-use success rate (where each turn is marked as either 0 or 1), then average the binary scores with the total dialogue turns of a dialogue session. This score takes into account the finer-grained success rate of tool-use at the turn level. (3) **Soft Averaged Turn Success Rate (SATS):** This metric adjusts the ATS based on the proximity of errors to the current turn. Specifically: If a turn is incorrect, the score is 0. If a turn is correct, given $j$ as the index of this turn and $i$ as the index of the last incorrect turn, the score is 1 when $j < i$ and $1 - e^{-(j-i)}$ when $j > i$. This design is based on the intuition that an incorrect earlier turn can negatively impact subsequent turns. Moreover, the earlier the turn becomes incorrect, the lower the overall accumulated score, even if the remaining turns are correct. (4) **Task Process Rate (TPR):** This is calculated as the ratio of the first incorrect turn to the total number of turns. This metric is included to capture how early in the dialogue the first mistake occurs, as earlier errors tend to disrupt the overall task flow more significantly.

**For Multi-Tool Scenarios (S-M & M-M).** For scenarios with multiple tools, the following metrics are used: (1) **Tool Number Accuracy (TN):** Denote the predicted tool list as "Pred" and the ground truth tool list as "GT", TN $=|$ Pred $\cap$ GT $|$ / $|$ Pred $\cup$ GT $|$. (2) **Tool Order Accuracy (TO):** Evaluate the correctness of the tool sequence, adjusted by a decay factor: TO $= t \times |$ LCR(GT, Pred) $|$ / $|$ GT $|$, where LCR is the longest common subsequence, and $t$ is a decay coefficient calculated as: $t = \cos\left((\pi/2) \times (i/|\text{Pred}|)\right)$, where $i$ is the starting position of the longest common subsequence. The value of $t$ ranges from 0 to 1, with a faster decay for positions later in the sequence.

Unlike conventional approaches that focus only on overall success rates (Qin et al., 2023b), our metrics account for the dynamics along dialogue turns and the dependencies between multiple tools. We refer the reader to Appendix D for more examples of how to compute these metrics.

## 3 Experiments

**Experimental Setup.** We evaluate 5 closed-source LLMs such as GPT-3.5 (**?**), GPT-4 (OpenAI, 2023), Qwen-Max (Team, 2024), GLM-4-Plus[1] and DeepSeek2.5 (DeepSeek-AI, 2024). We also evaluate numerous open-source LLMs such as LLaMA2 (Touvron et al., 2023) and LLaMA3 (AI@Meta, 2024) series, Qwen1.5 (Team, 2024) and Qwen2 (Yang et al., 2024) series, Mistral (Jiang et al., 2023), ChatGLM3 and GLM-4 (GLM et al., 2024; Du et al., 2022; Zeng et al., 2022) series, as well as . Then, we also compare 2 models specifically enhanced for tool-use: ToolLLaMA (Qin et al., 2023b) and our **MTU-LLaMA**, which is fine-tuned on MTU-Instruct based on LLaMA3-8B-Instruct. Note that all baselines are **instruction-tuned models**. We refer the reader to Appendix D for more details about the evaluation including the hard cases, the prompt templates, and the metric computation.

---

[1] https://bigmodel.cn/dev/api/normal-model/glm-4

Table 2: Results of different models on the **normal** set of MTU-Eval (**S-S & M-S**). "S-S" and "M-S" denote "Single-Turn Single-Tool" and "Multi-Turn Single-Tool" settings, respectively. We utilize green (1st), blue (2nd), and yellow (3rd) backgrounds to distinguish the top three results within both open-source and tool-use-specific models. We employ **bold** and underlined text to denote the top and second-best results across all model categories (same markers for the other tables). All the baselines are instruction-tuned models.

| Models | S-S | | | M-S | | | | | | |
|---|---|---|---|---|---|---|---|---|---|---|
| | TS | PS | Avg. | TS | PS | ATS | SATS | SR | TPR | Avg. |
| *Closed-Source Large Language Models* | | | | | | | | | | |
| GPT-4 | **95.83** | 52.08 | **73.96** | **88.10** | **74.49** | **73.67** | **67.36** | **29.63** | **45.35** | **63.10** |
| GPT-3.5 | 84.62 | 46.15 | 65.38 | 69.05 | 50.68 | 50.51 | 40.81 | 1.85 | 12.70 | 37.60 |
| Qwen-Max | 91.67 | 50.00 | 70.83 | 86.73 | 66.67 | 64.96 | 57.88 | 20.37 | 35.53 | 55.36 |
| GLM-4-Plus | **95.83** | 50.00 | 72.92 | 85.03 | 71.43 | 72.01 | 65.19 | 24.07 | 44.18 | 60.32 |
| DeepSeek V2.5 | 93.75 | 43.75 | 68.75 | 86.39 | 69.39 | 68.09 | 60.20 | 18.52 | 38.04 | 56.77 |
| *Open-Source Large Language Models* | | | | | | | | | | |
| LLaMA2-7B | 15.38 | 3.85 | 9.62 | 33.67 | 28.91 | 26.78 | 20.67 | 0.00 | 7.12 | 19.53 |
| LLaMA2-70B | 70.59 | 33.33 | 47.28 | 47.28 | 30.95 | 30.18 | 23.78 | 0.00 | 9.82 | 23.67 |
| LLaMA3-8B | 65.38 | 30.77 | 48.08 | 35.71 | 17.35 | 17.53 | 12.34 | 0.00 | 1.72 | 14.11 |
| LLaMA3-70B | 86.54 | **57.69** | 72.12 | 79.25 | 61.90 | 62.18 | 54.81 | 14.81 | 32.33 | 50.88 |
| Qwen1.5-14B | 75.00 | 34.62 | 54.81 | 62.93 | 36.73 | 35.17 | 27.37 | 1.85 | 6.34 | 28.40 |
| Qwen1.5-72B | 78.85 | 38.46 | 58.65 | 80.95 | 61.22 | 59.08 | 50.88 | 16.67 | 28.75 | 48.75 |
| Qwen2-7B | 73.08 | 38.46 | 55.77 | 71.09 | 49.66 | 49.59 | 40.14 | 5.56 | 13.30 | 38.22 |
| Qwen2-72B | 86.54 | 48.08 | 67.31 | 79.93 | 61.22 | 58.52 | 50.28 | 14.81 | 25.15 | 48.32 |
| Mistral-7B | 60.42 | 25.00 | 42.71 | 61.22 | 37.07 | 37.33 | 29.14 | 3.70 | 9.98 | 29.74 |
| ChatGLM3-6B | 10.00 | 0.00 | 5.00 | 22.90 | 5.99 | 5.79 | 3.66 | 0.00 | 0.00 | 6.39 |
| GLM-4-9B | 91.67 | 45.83 | 68.75 | 63.95 | 42.18 | 42.72 | 35.98 | 3.70 | 19.01 | 34.59 |
| *Tool-Use-Specific Large Language Models* | | | | | | | | | | |
| ToolLLaMA2-7B | 85.42 | 18.75 | 52.08 | 31.97 | 7.82 | 7.6 | 5.20 | 0.0 | 5.73 | 9.72 |
| MTU-LLaMA (Ours) | 92.31 | 50.00 | 71.15 | 81.63 | 67.69 | 66.94 | 58.74 | 9.26 | 32.47 | 52.79 |

## 3.1 MAIN RESULTS

**Overall Performance.** The experimental results for the normal set are presented in Table 2 (S-S & M-S) and Table 3 (S-M & M-M). The results on the hard set are illustrated in Table 4. These results reveal several key findings: (1) Open-source models typically exhibit inferior performance compared to closed-source models in nearly all metrics, with the exception of GPT-3.5. However, certain models, including LLaMA3-70B and Qwen2-72B, demonstrate results comparable to those achieved by closed-source models. (2) GPT-4 consistently exhibits superior performance on the normal set; however, its performance decreases on the hard set compared to GLM-4-Plus. Qwen-Max demonstrates exceptional performance in the M-M setting, with its advantages becoming more pronounced in the hard setting, even surpassing GPT-4. Similarly, GLM-4-Plus exhibits outstanding performance in the S-S and M-S settings, and its superiority is further amplified in the hard setting, also exceeding that of GPT-4. DeepSeek V2.5 performs admirably in the S-M setting. (3) Our MTU-LLaMA exhibits substantial advancements over its initialization, *i.e.*, LLaMA3-8B-Instruct, across all settings and metrics. It is also competitive with some closed-source models, underscoring the effectiveness of our MTU-Instruct. (4) Generally, all the models perform better on the normal set than on the hard set, indicating LLMs' limitations in handling more challenging tool-use scenarios. (5) Notably, despite being fine-tuned specifically for tool-use, ToolLLaMA exhibits poor performance across all settings and metrics, suggesting its limited generalizability.

**Effect of Multi-Turn.** We compare the single-turn (S-S, S-M) and multi-turn (M-S, M-M) settings across the Tables 2, 3, and 4, and have following findings: (1) Both closed-source and open-source models tend to perform worse in multi-turn settings (M-S and M-M) compared to single-turn settings (S-S and S-M). (2) Our MTU-LLaMA shows relatively better adaptation and robustness to multi-turn settings. (3) Based on our novel TPR metric, we can observe that LLMs typically experience tool-use errors within the initial 30%-50% turns for closed-source models, and within the first 0%-30% turns for open-source models. (4) Most models such as Qwen2-72B have significantly higher ATS scores than TPR scores. This implies that while LLMs frequently encounter tool-use errors in the initial

Table 3: Results of different models on the **normal** set of MTU-Eval (**S-M & M-M**). "S-M" and "M-M" denote "Single-Turn Multi-Tool" and "Multi-Turn Multi-Tool" settings, respectively.

| Models | S-M | | | M-M | | | | | | |
|---|---|---|---|---|---|---|---|---|---|---|
| | TN | TO | Avg. | TN | TO | ATS | SATS | SR | TPR | Avg. |
| *Closed-Source Large Language Models* | | | | | | | | | | |
| GPT-4 | **66.85** | **70.52** | **68.68** | **72.10** | **73.38** | **68.77** | **66.07** | **30.95** | **59.52** | **61.80** |
| GPT-3.5 | 32.64 | 38.22 | 35.43 | 24.72 | 25.46 | 16.11 | 12.22 | 0.00 | 3.97 | 13.75 |
| Qwen-Max | 39.76 | 48.82 | 39.29 | 62.00 | 64.07 | 56.27 | 55.27 | 4.76 | 52.38 | 49.13 |
| GLM-4-Plus | 45.76 | 48.48 | 47.12 | 53.95 | 54.58 | 49.17 | 45.72 | 4.76 | 39.48 | 41.28 |
| DeepSeek V2.5 | 56.88 | 60.28 | 58.58 | 50.15 | 51.79 | 44.84 | 41.26 | 7.14 | 34.72 | 38.32 |
| *Open-Source Large Language Models* | | | | | | | | | | |
| LLaMA2-7B | 3.39 | 3.94 | 3.67 | 22.22 | 22.22 | 22.90 | 21.80 | 0.00 | 19.92 | 19.92 |
| LLaMA2-70B | 6.82 | 8.48 | 7.65 | 30.12 | 30.49 | 28.77 | 28.77 | 0.00 | 28.77 | 28.77 |
| LLaMA3-8B | 14.79 | 20.30 | 17.55 | 9.43 | 10.04 | 4.44 | 2.81 | 0.00 | 0.00 | 4.46 |
| LLaMA3-70B | 26.85 | 32.68 | 29.76 | 33.60 | 35.71 | 26.94 | 23.82 | 0.00 | 17.86 | 22.99 |
| Qwen1.5-14B | 22.12 | 28.22 | 25.17 | 27.78 | 28.67 | 21.07 | 19.04 | 0.00 | 14.88 | 18.57 |
| Qwen1.5-72B | 28.04 | 30.60 | 29.32 | 23.00 | 23.31 | 52.94 | 51.86 | 0.00 | 7.34 | 26.41 |
| Qwen2-7B | 24.52 | 29.59 | 27.05 | 21.04 | 22.75 | 15.24 | 11.52 | 0.00 | 4.76 | 12.55 |
| Qwen2-72B | 52.76 | 59.98 | 56.37 | 45.93 | 47.67 | 42.02 | 38.07 | 7.14 | 29.76 | 29.76 |
| Mistral-7B | 14.21 | 18.22 | 16.22 | 10.15 | 11.11 | 5.44 | 3.66 | 0.00 | 0.60 | 5.16 |
| ChatGLM3-6B | 6.53 | 8.56 | 7.55 | 10.64 | 11.11 | 9.21 | 8.01 | 2.38 | 5.95 | 7.88 |
| GLM-4-9B | 23.64 | 27.58 | 25.61 | 16.17 | 16.45 | 9.48 | 6.13 | 0.00 | 0.00 | 8.04 |
| *Tool-Use-Specific Large Language Models* | | | | | | | | | | |
| ToolLLaMA2-7B | 11.52 | 11.52 | 11.51 | 4.07 | 4.07 | 2.78 | 2.34 | 0.00 | 1.59 | 2.48 |
| MTU-LLaMA (Ours) | 55.39 | 58.55 | 56.97 | 42.47 | 43.42 | 39.64 | 32.50 | 7.14 | 19.05 | 30.70 |

turns, they can still correctly use tools in subsequent turns in most cases. However, this correctness does not account for the cascading effect of previous errors, but solely considers the success rate of independent tool usage. (5) Fortunately, the SATS scores can be treated as an equilibrium between ATS and TPR metrics, which simultaneously account for the positions at which tool-use errors occur and the subsequent impact on later turns.

**Effect of Multi-Tool.** Based on multi-tool settings (S-M and M-M) across Tables 2, 3, and 4, we derive the following findings: (1) Multi-tool settings (S-M and M-M) show significant complexity, leading to noticeable performance drops for most models. (2) Despite the complexity, models like GPT-4, Qwen2-72B and our MTU-LLaMA show stronger robustness. (3) In contrast, the good models in single-tool settings such as GLM-4-Plus (closed-source) and LLaMA3-70B (open-source), are surpassed by Qwen-Max (closed-source) and Qwen2-72B (open-source), respectively, indicating the superior performance of Qwen series in multi-tool settings. (4) The model rankings by TN and TO are highly consistent, implying that models with better control over the number of tools also tend to manage tool sequences effectively, suggesting a strong correlation between these capabilities.

## 3.2 ANALYSIS

**OOD Performance.** To evaluate the generality of MTU-LLaMA, we measure its performance on the OOD test split of MTU-Bench and two other OOD tool-use benchmarks, *i.e.*, API-Bank (Li et al., 2023) and ToolTalk (Farn & Shin, 2023). in Table 5, we com-

Table 4: Results on the **hard** set of MTU-Eval. We report the average scores for each setting (Detailed results are shown in Appendix E).

| Models | S-S | M-S | S-M | M-M |
|---|---|---|---|---|
| *Closed-Source Large Language Models* | | | | |
| GPT-4 | 77.88 | 44.61 | **58.07** | 41.36 |
| GPT-3.5 | 41.96 | 30.86 | 18.39 | 11.87 |
| Qwen-Max | 77.88 | 42.11 | 24.01 | **45.08** |
| GLM-4-Plus | **82.69** | **47.61** | 30.90 | 39.53 |
| DeepSeek V2.5 | 80.77 | 44.94 | 40.01 | 30.62 |
| *Open-Source Large Language Models* | | | | |
| LLaMA2-7B | 28.57 | 17.13 | 2.35 | 11.76 |
| LLaMA2-70B | 28.57 | 23.46 | 1.74 | 16.79 |
| LLaMA3-8B | 25.89 | 12.85 | 9.91 | 5.89 |
| LLaMA3-70B | 71.43 | 40.40 | 20.67 | 20.56 |
| Qwen1.5-14B | 44.64 | 29.39 | 12.81 | 9.37 |
| Qwen1.5-72B | 56.73 | 29.92 | 18.85 | 17.93 |
| Qwen2-7B | 58.93 | 28.73 | 17.50 | 10.17 |
| Qwen2-72B | 68.40 | 38.42 | 37.14 | 25.13 |
| Mistral-7B | 26.92 | 26.04 | 11.48 | 10.84 |
| ChatGLM3-6B | 9.09 | 5.57 | 18.89 | 9.52 |
| GLM-4-9B | 47.12 | 30.22 | 18.98 | 9.52 |
| *Tool-Use-Specific Large Language Models* | | | | |
| ToolLLaMA2-7B | 18.27 | 10.19 | 0.51 | 2.34 |
| MTU-LLaMA (Ours) | 37.5 | 43.10 | 39.31 | 24.70 |

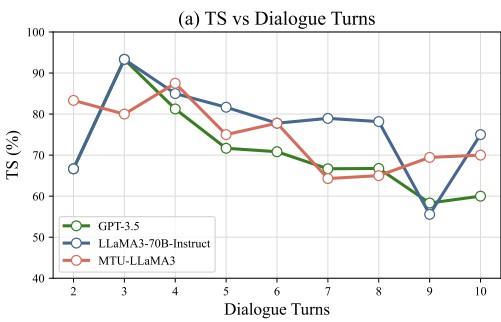 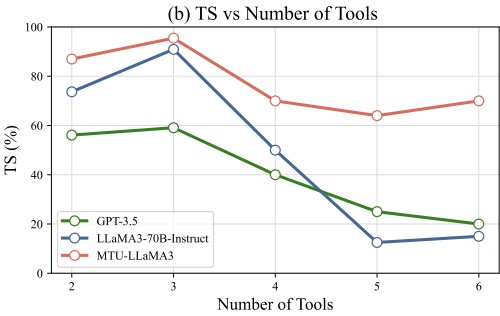

Figure 5: (a) Effect of dialogue turns. (b) Effect of different number of tools.

pare the performance of MTU-LLaMA, LLaMA3-8B-Instruct, and GPT-4 on these benchmarks under the M-S setting, and We observe that MTU-LLaMA outperforms LLaMA3-8B-Instruct on all three OOD benchmarks. Notably, MTU-LLaMA achieves performance comparable to that of GPT-4 on API-Bank, which show strong generalizability of MTU-LLaMA.

**Effect of Dialogue Turns.** We illustrate the impact of dialogue turns on tool selection accuracy in Figure 5. We observe that performance slightly declines as the number of dialogue turns increases. Our MTU-LLaMA exhibits the most gradual decrease in performance, demonstrating its robustness to higher dialogue turns.

**Effect of Tool Numbers.** Figure 5 shows the impact of tool numbers on tool selection accuracy. As the number of tools increases, the performance of GPT-3.5 and LLaMA3-70B degrades, with LLaMA3-70B showing a sharper drop. In contrast, MTU-LLaMA maintains relatively stable accuracy, demonstrating its superior handling of multiple tool calls.

Table 5: OOD Performance of our MTU-LLaMA.

| Models | MTU-Bench (OOD) | | | | | | |
|---|---|---|---|---|---|---|---|
| | TS | PS | ATS | SATS | SR | TPR | Avg. |
| GPT-4 | 67.28 | 67.57 | 65.19 | 47.23 | 32.31 | 37.72 | 52.88 |
| LLaMA3-8B | 36.71 | 36.71 | 35.53 | 23.00 | 7.69 | 11.31 | 25.16 |
| MTU-LLaMA | 47.34 | 37.88 | 62.05 | 54.58 | 9.26 | 35.49 | 41.10 |
| Models | ToolTalk | | | | | | |
| | TS | PS | ATS | SATS | SR | TPR | Avg. |
| GPT-4 | 51.08 | 51.60 | 44.90 | 39.74 | 6.90 | 27.13 | 36.89 |
| LLaMA3-8B | 30.00 | 30.86 | 26.72 | 22.66 | 0.00 | 15.44 | 20.95 |
| MTU-LLaMA | 30.25 | 31.23 | 30.87 | 28.51 | 3.45 | 22.82 | 24.52 |
| Models | API-Bank | | | | | | |
| | TS | PS | ATS | SATS | SR | TPR | Avg. |
| GPT-4 | 48.56 | 48.56 | 45.56 | 44.59 | 38.32 | 41.10 | 44.45 |
| LLaMA3-8B | 14.03 | 14.03 | 13.86 | 11.94 | 7.74 | 8.63 | 11.71 |
| MTU-LLaMA | 51.80 | 51.80 | 48.58 | 45.57 | 38.10 | 38.10 | 45.66 |

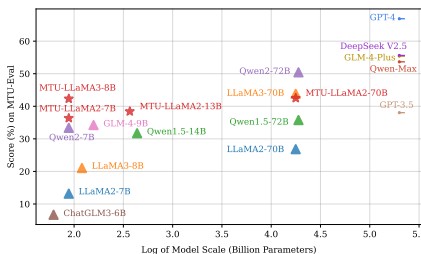

Figure 6: Scaling Law of LLMs on MTU-Eval.

**Error Analysis.** In Figure 7, we use five LLMs to analyze the different types of errors (i.e., "Action Error", "Parameter Error" and "Format Error"). Specifically, the "Action Error" and "Parameter Error" denotes to select wrong tools and wrong parameters, respectively, and the "Format Error" means that the model cannot follow instructions well and outputs wrong formats, which cannot be resolved well. Figure 7 illustrates two primary findings: (1) "Action Error" occurs more often than other errors, specifically in challenge M-M setting, and stronger models show fewer errors. (2) "Format Error" usually exists in weaker LLMs (e.g., LLaMA3-8B-Instruct and ChatGLM4-9B), and our fine-tuned version MTU-LLaMA greatly reduces format issues, which shows the effectiveness of MTU-Instruct. See Appendix E for detailed error cases.

**Scaling Law** We evaluate the performance of MTU-LLaMA across different model sizes, using LLaMA2 models with 7B, 13B, and 70B parameters as initialization, which are fine-tuned with MTU-Instruct. The results in Figure 6 show that the performance of MTU-LLaMA improves as the model size increases, suggesting its scalability.

**Consistency Between Our Proposed Metrics and Human Evaluation.** To show the validity of our proposed metrics (SATS, TN, and TO), we evaluate the consistency between these novel metrics and the human evaluation results. We randomly sample 50 instances from the M-S subset for the

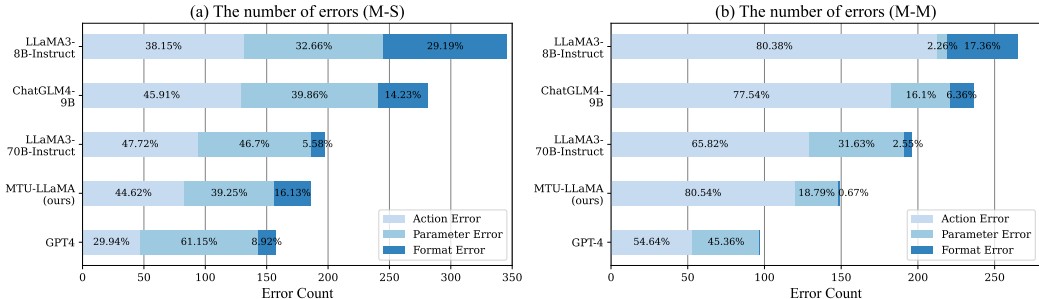

Figure 7: Analysis of different error types for different LLMs under M-S and M-M settings.

SATS metric and 50 instances from the S-M subset for the TN and TO metrics to compare two models: GPT-3.5 and LLaMA3-8B. Human annotators are also asked to compare the two models. In Table 6, we report the Pearson correlation coefficient between our metrics and human evaluation results, as well as the Pearson correlation among human annotators, which shows excellent consistency between these metrics and human evaluations. More details can be found in Appendix E.

## 4 RELATED WORKS

**Instruction Tuning for Tool Learning.** The objective of tool learning is to equip large language models (LLMs) with human-like tool usage capabilities (Yang et al., 2023; Shen et al., 2023; Qu et al., 2024). To achieve this, LLMs are typically fine-tuned with tool instruction data to improve their performance in tool planning, selection, calling, and response generation (Schick et al., 2023; Liang et al.,

Table 6: Consistency for our proposed metrics.

| Metric | Annotation Count | Consistency | |
| --- | --- | --- | --- |
| | | Metric-Human | Human-Human |
| SATS | 50 | 0.8280 | 0.9344 |
| TN | 50 | 0.8497 | 0.9245 |
| TO | 50 | 0.8821 | 0.9482 |

2023; Kong et al., 2023). However, existing tool instruction datasets either have limitations in multi-turn dialogue and multi-tool usage scenarios, or are based on synthetic data, resulting in misalignment with real-world user needs (Huang et al., 2024; Li et al., 2023; Patil et al., 2023; Zhuang et al., 2023). In this paper, we introduce MTU-Instruct, a large-scale instruction dataset to improve LLMs' performance in diverse real-world tool-use scenarios.

**Evaluation Benchmarks for Tool Learning.** Many tool-use benchmarks have been proposed, but they still have many limitations. Firstly, in Table 1, these benchmarks have limited capabilities to assess complex scenarios (e.g., multi-turn dialogues, multiple tools, and cross-domain tool generalization) (Huang et al., 2024; Li et al., 2023; Patil et al., 2023; Tang et al., 2023). Secondly, some benchmarks rely excessively on GPT models, potentially leading to subjective and unstable results with heavy costs (Qin et al., 2023b; Tang et al., 2023). Finally, existing assessments often overlook critical dimensions, such as the order of multi-tool invocation, the impact of erroneous calls on subsequent interactions, and the accuracy of tool parameter selection, resulting in evaluations lacking comprehensiveness and depth (Zhuang et al., 2023; Li et al., 2023; Patil et al., 2023). In contrast, MTU-Bench not only includes extensive multi-turn dialogues and multiple tool scenarios but also introduces testing for OOD tool generalization. By employing automated evaluation and incorporating metrics like SATS, TN, and TO, MTU-Eval achieves a more comprehensive assessment.

## 5 CONCLUSION

In this work, we propose a multi-granularity tool-use benchmark for LLMs called **MTU-Bench**, which consists of MTU-Instruct and MTU-Eval. Specifically, first, the MTU-Instruct dataset is used to enhance the tool-use abilities of existing LLMs, and the MTU-Eval with multiple tool-use scenes is applied to benchmark the tool-use abilities comprehensively. Notably, all evaluation metrics of our MTU-Eval are based on the prediction results and the ground truth without using any GPT or human evaluation metrics. Moreover, Comprehensive experimental results demonstrate the effectiveness of our MTU-Bench. Finally, we hope MTU-Bench can guide developers and researchers in understanding the tool-use capabilities of LLMs and facilitate the growth of foundation models.

ETHICS

In developing MTU-Bench and MTU-LLaMA, we recognize several ethical considerations that arise from the broader context of integrating tool-use capabilities into large language models (LLMs). As these models become more capable of interacting with real-world systems—such as those involving financial services, healthcare, and other critical domains—we must consider the potential risks associated with misuse. For instance, there is the possibility that LLMs could be exploited to access sensitive tools or manipulate information in ways that could harm individuals or organizations.

While our work aims to improve the accuracy and efficiency of tool use, we are mindful of the importance of ensuring that these technologies are deployed responsibly. We advocate for the implementation of robust safeguards, including transparency in decision-making processes, fairness in how tools are applied, and accountability in real-world usage. Furthermore, we encourage future research and development efforts to focus on mitigating potential biases and ensuring that these systems are secure and trustworthy when handling sensitive tasks.

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

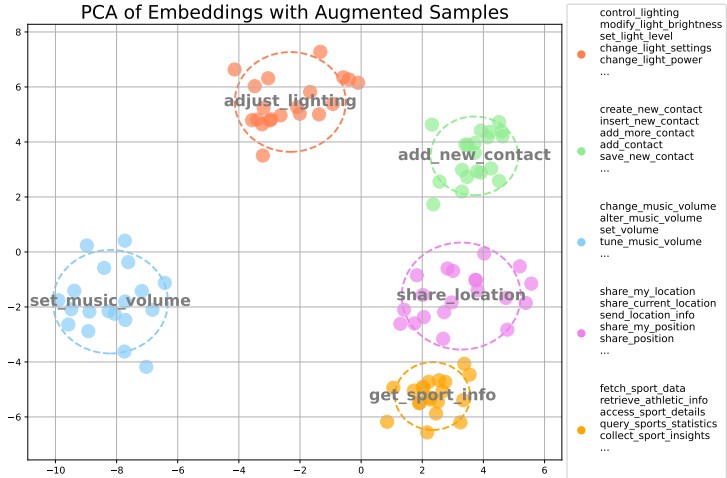

Figure 8: We extract embeddings for the names of tools and apply a fixed distance threshold for clustering. In this Figure, we present the clustering visualization results after PCA for partial samples from five categories.

## A    LIMITATIONS

Despite the strengths of our proposed MTU-Bench, there are still several limitations to consider. First, although MTU-Bench incorporates a diverse range of real-world user instructions, it may not fully capture all potential edge cases or highly complex interactions that can occur in dynamic, real-world environments. Second, while MTU-Eval provides comprehensive fine-grained metrics for evaluating tool-use abilities, these metrics are based on predefined benchmarks, which may not account for every possible tool-use challenge in evolving real-world applications. Although MTU-LLaMA demonstrates strong generalization across various metrics and scenarios, further research is needed to explore its adaptability to increasingly complex and emerging tool functionalities in real-time settings. Future work should focus on broadening the benchmark's coverage and exploring more dynamic and complex real-world use cases.

## B    DETAILS OF MTU-BENCH CONSTRUCTION

**Details for Data Filtering.**    We perform a rigorous data filtering process to ensure that the data selected for MUT-Bench is reliable and of high quality. The filtering process is as follows:

- **Exclusion of Unsuitable Intents:** We filter out intents that are not suitable for tool calls, particularly those that are difficult to tackle with external tools. For example, conversations seeking naming suggestions are excluded, as they are inherently challenging to define for tool usage. Manual verification is employed to achieve this, resulting in a 26% data exclusion rate.

- **Redundancy Elimination through Clustering:**    The synthesis of tools can lead to redundancies, such as `share_location`, `share_my_location`, and `send_location_info`. To mitigate this, we adopt a clustering approach to group similar tools, retaining only one representative tool from each cluster. We establish a lower threshold for initial clustering to ensure each cluster contains a unique tool, followed by manual elimination of any duplicates. This process achieves a reduction ratio of approximately 20:1, as illustrated in Figure 8.

- **Filtering of Undefined Tools:** About 6% of the synthesized data includes tools that are not defined in the tool library. This data is filtered out using rule-based matching methods.

- **Parameter Correctness Check:** Approximately 16.9% of the synthesized data fails our correctness checks, comprising 3.2% of cases with fabricated non-existent parameters and 13.7% where generated parameters do not meet format validation requirements.

- **LLM Verification:** We utilize an LLM, specifically GPT-4, to recheck the correctness of all answers. This process results in an additional 10% of data being filtered out.

- **Manual Quality Annotation:** We conduct manual quality checks utilizing multiple experts. For the training set, 500 samples are randomly selected and each is evaluated by three experts, with any discrepancies resolved by a fourth. This approach yields a 96% accuracy rate. For the test set, all samples are meticulously calibrated by hired experts, achieving a final accuracy of 100%.

**Prompt Templates for MTU-Bench Construction.** The construction procedure of MTU-Bench involves numerous prompt templates, as listed below, for tool making (tool name synthesis and tool parameter synthesis), thought synthesis, observation simulation, and data quality check.

---

**Prompt Template (Tool Name Synthesis)**

I will give you a dialogue, including historical dialogue information and current round dialogue information. Please help me determine whether Assistant needs to call an \"API\" to obtain specific information or perform certain operations in order to solve the user issue in the current round of conversation.
1. If yes, and Assistant's response shows that the user's query has been solved, then return me with \"api_name\" in \"tags\". The name of the API should be concise and easy to understand, such as search_restaurant, book_restaurant, etc. The API name should start with a verb and include the specific domain name. Related domains includes [domain_info ], etc. Note that if multiple APIs need to be called, return me with \"api_name, api_name, ...\" in \"tags\".
2. If Assistant can provide the current response without calling the API, return me \"no need to call\"in \"tags\".
3. If Assistant needs to call an API but is unable to do so due to a lack of necessary information, inform me of \"lack of necessary information\" in \"tags\". For example, due to the lack of "restaurant name", the API for "book_restaurant" cannot be called.
4. If Assistant is only confirming existing information with the User, inform me of the \"confirmation information\" in \"tags\".
6. If some information in the historical conversation can help Assistant respond to the current issue without calling APIs, inform me that \"information already exists\" in \"tags\".
Please reply to me in JSON format: {\"Analysis\": str, \"tags\": str}.

---

**Prompt Template (Tool Parameter Synthesis)**

I will provide you with a conversation segment that includes both historical dialogue information and current round dialogue information. Based on this information, please help me determine whether, to solve the problem presented by the User in the current round of dialogue, the Assistant calls a specific API to obtain the necessary information or to perform related actions. Please respond according to the following guidelines:
1. If an API call is required and the Assistant's response solves the User's problem, please specify which API was called and reply to me in the following JSON format: \"Action\": \"api_name\". Also, provide the parameters required for calling that API in the format: \"Action Input\": {\"parameter_name\": \"value\", ...}".
2. If it is impossible to call an API due to missing necessary parameter information, please explain in the \"Thought\" section due to the absence of which parameters, which API cannot be called.
3. If answering the User's question does not require calling an API, please explain in the \"Thought\" section why there is no need to use a API.
4. Please strictly use the API names and parameter names I provide, and refrain from fabricating any. If the required parameter is not defined within our list, you are allowed to introduce new parameter names. Beyond this allowance, do not utilize any API names and parameter names that are beyond the specified range, to ensure consistency and accuracy.
5. Please include a section called \"Thought\" in your answer where you clearly and unambiguously demonstrate your thought process when solving or answering the question.
Below are the APIs and their parameters information you can use:
[apis_information]
Please reply in the following JSON format: {\"Thought\": \"str\", \"Action\": \"api_name\", \"Action Input\": {\"parameter_name\": \"value\", ...}}. If there is no information for \"Action\", \"Action Input\", or \"Thought\", please return an empty string.

---

**Prompt Template (Thought Synthesis)**

I will provide you with a multi−round conversation and API call information for each round. Your task is to add or verify the Assistants' thinking process of API call to the dialog.

Your need to return the modified conversation with API call results added to me, strictly following the following format:

\nModified Dialogue:\n[Round 0]\nUser: xxx.\nThought: xxx.\nAction: xxx.\nAction Input : xxx.\nObservation: xxx.\nAssistant: xxxx. \n[Round 1]\nUser: xxx.\nThought: xxx.\nAssistant: xxx.\n[Round 2]...

That is to say, after each round of the user's question, the Assistant must think about whether to call the API and how to call it, and present this in \"Thought\".

For example:

Modified Dialogue:

[Round 0]

User: Find me some fantasy movies I can watch online, something directed by Neil Marshall.

Thought: The user is looking for fantasy movies directed by Neil Marshall to watch online. I need to query a database or service for movies that fit these criteria.

Action: FindMovies

Action Input: {'directed_by': 'Neil Marshall', 'genre': 'Fantasy'}

Observation: {'directed_by': 'Neil Marshall', 'genre': 'Fantasy', 'subtitles': 'False', 'title': 'Hellboy'}

Assistant: Are you interested in Hellboy?

[Round 1]

User: I would enjoy watching Hellboy.

Thought: The user has not expressed any request, so the tool is not invoked here.

Assistant: Shall I play the movie?

[Round 2]

User: Yes, begin playing the movie. Yes, play it without subtitles.

Thought: The user has confirmed they want to start watching the movie "Hellboy" and specifically requested to play it without subtitles. I'll proceed to execute the play command with subtitles disabled.

Action: PlayMovie

Action Input: {'subtitles': 'False', 'title': 'Hellboy'}

Observation: {'directed_by': 'Neil Marshall', 'genre': 'Fantasy', 'subtitles': 'False', 'title': 'Hellboy'}

Assistant: The movie is now playing.

Requirements:

1. After \"Action\", it shows which API to call. If not, it indicates that API calls are not required for this round. The parameters passed in by calling the API are displayed after \"Action Input\". You just need to supplement the API call return results for the rounds that have called the API, and place them after \"Observation\". \"Observation\" shows the return result of the API.

2. Please return the thinking process in \"Thought\".

5. I will provide you with reference APIs and parameter names.

6. Except for adding \"Thought\" to the round of API calls, all other information remains in its original content.

7. Please return the complete conversation.

The following is a list of APIs and their parameters for your reference:

[apis information]

---

> **Prompt Template (Observation Simulation)**
>
> I will provide you with a multi−round conversation and API call information for each round. Your task is to add the return result of an API call to the dialog based on Assistant's response.
> You need to return the modified conversation with API call results added to me, strictly following the following format:
> \nModified Dialogue:\n[Round 0]\nUser: xxx.\nThought: xxx.\nAction: xxx.\nAction Input : xxx.\nObservation: xxx.\nAssistant: xxxx. \n[Round 1]\nUser: xxx.\nThought: xxx.\ nAssistant: xxx.\n[Round 2]...
> For example:
> Modified Dialogue:
> [Round 0]
> User: Find me some fantasy movies I can watch online, something directed by Neil Marshall.
> Thought: The user is looking for fantasy movies directed by Neil Marshall to watch online. I need to query a database or service for movies that fit these criteria.
> Action: FindMovies
> Action Input: {'directed_by': 'Neil Marshall', 'genre': 'Fantasy'}
> Observation: {'directed_by': 'Neil Marshall', 'genre': 'Fantasy', 'subtitles': 'False', 'title': 'Hellboy'}
> Assistant: Are you interested in Hellboy?
> [Round 1]
> User: I would enjoy watching Hellboy.
> Assistant: Shall I play the movie?
> [Round 2]
> User: Yes, begin playing the movie. Yes, play it without subtitles.
> Thought: The user has confirmed they want to start watching the movie "Hellboy" and specifically requested to play it without subtitles. I'll proceed to execute the play command with subtitles disabled.
> Action: PlayMovie
> Action Input: {'subtitles': 'False', 'title': 'Hellboy'}
> Observation: {'directed_by': 'Neil Marshall', 'genre': 'Fantasy', 'subtitles': 'False', 'title': 'Hellboy'}
> Assistant: The movie is now playing.
> Requirements:
> 1. After \"Action\", it shows which API to call. If not, it indicates that API calls are not required for this round. The parameters passed in by calling the API are displayed after \" Action Input\". You just need to supplement the API call return results for the rounds that have called the API, and place them after \"Observation\".
> 2. Please return the information in \"Observation\" in JSON format, for example: {\" parameter_name\": \"value\", \"parametername\": \"value\"...}. Specifically, in every round with \"Action\" and \"Action Input\", you should add an \"Observation\" after the \"Action Input\", which should fill in the information returned by the API. For example: \nAction: FindMovies \nAction Input: {'directed_by': 'Neil Marshall', 'genre': 'Fantasy'}\ nObservation: {'directed_by': 'Neil Marshall', 'genre': 'Fantasy', 'subtitles': 'False', 'title ': 'Hellboy'}.
> 3. If Assistant's response shows that it has not yet received the specific information returned by the API tool call, that is "\nObservation\": {\"error\": \"Time out.\"}
> 4. If the result returned by Assistant shows that no relevant information is found, the API call returns an empty dict, such as: "\nObservation\": {}.
> 5. I will provide you with reference APIs and parameter names. Ensure that all parameter names used are defined and should not be fabricated.
> 6. Except for adding API return results to the round of API calls, all other information remains in its original content.
> 7. Please return the complete conversation.
> The following is a list of APIs and their parameters for your reference:
> [apis information]

---

**Prompt Template (Data Quality Check)**

Please review the provided conversation snippet, which includes historical dialogue, current round dialogue, and the API call made in this round. Your task is to verify the accuracy of the API and parameters used in this round of dialogue. Assume the Assistant does not have knowledge of real−world information such as cinemas or restaurants; it relies on API calls to access information or carry out actions such as making a reservation. Use the guidelines below to correct any inaccuracies.
1. Check and correct the API selection in the \"Action\" field. Common errors include: a. Assistant's response indicates that an API call was made, but the \"Action\" field is empty.
b. Assistant don't need to call any API to reply to the user's current round of conversation. In this case, calling is not necessary, but there is an API name in the \"Action\" field. c. The assistant's response shows that the necessary information required for API calls is missing. Therefore, the assistant is asking the user for additional information, but there is an API name in the \"Action\". d. The API listed in the \"Action\" is incorrect.
2. Verify and correct the parameters listed in \"Action Input\" and ensure they correctly match the API call. The \"Action Input\" should be formatted as {\"parameter_name\": \"value\", \"parameter_name\": \"value\", ...}.
3. Revise the content in \"Thought\" to include the correct rationale for selecting tools and parameters. The thought should only consider historical conversations and current user issues, assuming that the assistant's response is unknown.
4. Ensure all API and parameter names used are as defined and should not be invented. Here is a list of APIs and their parameters for your reference:
[apis_information]
Please respond in the following JSON format: {\"Thought\": str, \"Action\": \"api_name\", \"Action Input\": {\"parameter_name\": \"value\", ...}}. If there is no \"Action\" or \"Action Input\" information, please return an empty string.

---

**Example of Tool Document.** As shown in Figure 9, the tool document allows the model to determine the appropriate tool names and their usages. It contains all the tools we synthesized, each tool including its corresponding tool description, necessary parameters, optional parameters, parameter description and data type, as well as the returns.

## C    MORE MTU-BENCH DATA ANALYSIS.

**Length Distribution.** We illustrate the length distributions in Figure 10, and Figure 11 for the training and evaluation data, respectively.

**More Statistics.** Table 7 provides a comprehensive summary of the dialogue statistics across diverse settings, distinct splits and subsets.

## D    DETAILS OF MTU-EVAL

**Hard cases in the hard test set.** encompass extensive parameters, nonsensical tool names, determination of specific parameter values, inability to call tools, interaction among multiple tools, and multi-turn parameter inheritance, as delineated in Table 8.

**Prompt Templates for MTU-Eval.** During the evaluation, the models are provided with distinct system prompts for various settings, which encompass both the comprehensive task specifications and the tool documentation. We list the system prompts used for evaluation in Boxes D.

Table 7: The number of dialogues under different settings.

| Setting | Train | Test | |
|---|---|---|---|
| | | normal | hard |
| **S-S** | 14277 | 52 | 56 |
| **S-M** | 13641 | 55 | 39 |
| **M-S** | 19007 | 54 | 31 |
| **M-M** | 7442 | 42 | 37 |
| **OOD** | - | 65 | |

```
{
   "name": "setAlarm",
   "description": "This tool is used for setting a new alarm based on the user's specified time,
label, recurrence pattern, sound preference, and specific day(s) for the alarm to activate.",
   "required_parameters": [
      {
         "name": "time",
         "type": "string",
         "description": "The specified time for the alarm to go off.",
         "format": "HH:MM"
      },
      {
         "name": "label",
         "type": "string",
         "description": "A custom name or description for the alarm.",
      }
   ],
   "optional_parameters": [
      {
         "name": "recurrence",
         "type": "array",
         "description": "It specifies how often the alarm should repeat. Each element in the list can
be a keyword ('everyday', 'weekdays', 'weekends'), a name of the day ('Monday', 'Tuesday',
'Wednesday', 'Thursday', 'Friday', 'Saturday', 'Sunday'), or a specific date in 'YYYY-MM-DD'
format.",
         "items": {
            "type": "string",
            "anyOf": [
               {"enum": [ "everyday", "weekdays", "weekends", "Monday", "Tuesday",
"Wednesday", "Thursday", "Friday", "Saturday", "Sunday" ]},
               {"pattern": "^\\d{4}-\\d{2}-\\d{2}$"} ],
            "default": ["everyday"]}
      },
      {
         "name": "sound",
         "type": "string",
         "description": "The chosen sound for the alarm when it goes off."
      },
      {
         "name": "vibrate",
         "type": "int",
         "description": "Specifies the intensity of the vibration for the alarm. A value of 0 means
no vibration. If the intensity of the vibration is not specified, the default intensity is 5.",
         "default": 0
      }
   ],
   "result_parameters": []
}
```

Figure 9: The JSON structure of the tool "setAlarm", which is used to create a new alarm based on user-specified parameters. The structure includes required, optional and result (*i.e.*, return) parameters, along with their corresponding data types, descriptions, formats and default values.

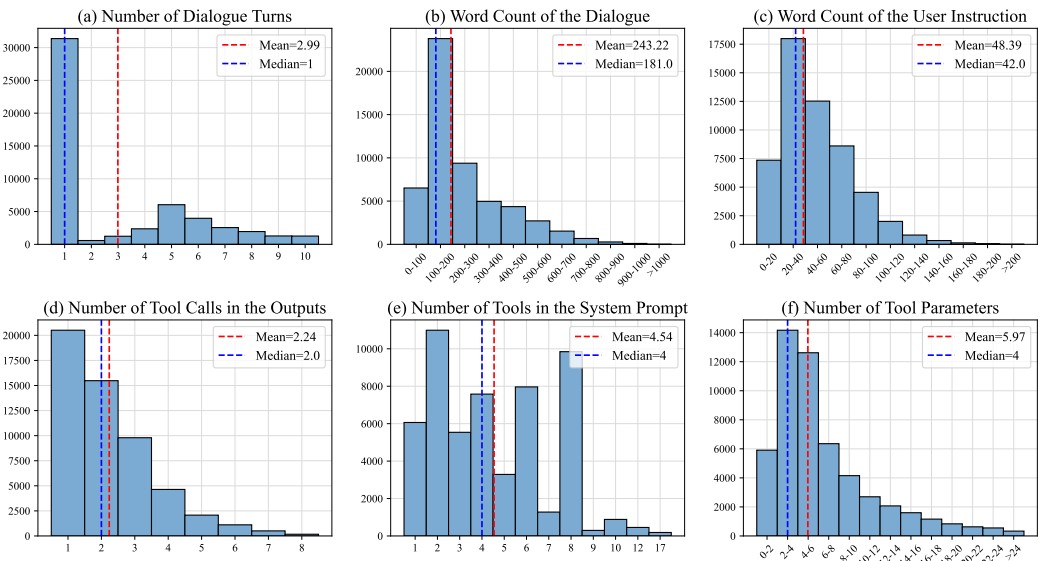

Figure 10: Length distributions of the training data.

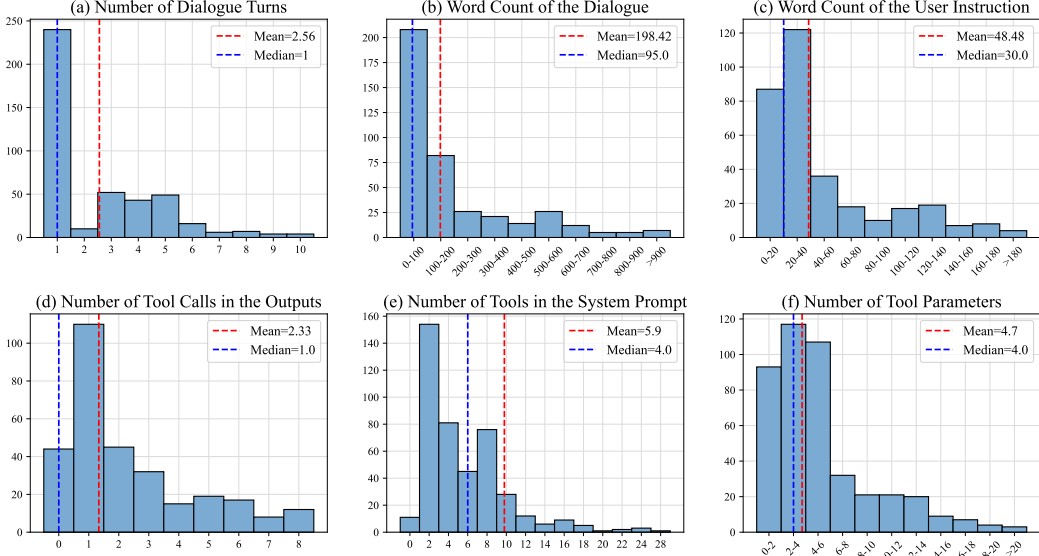

Figure 11: Length distributions of the evaluation data.

Table 8: Hard cases in the hard test set.

| Type | Description | Case |
|---|---|---|
| Extensive Parameters | The quantity of parameters required to be specified exceeds six. | User: Can you book me a non-refundable, one-stop flight on Delta Airlines from New York to Chicago, leaving on the 5th and returning on the 11th of this month in Economy class?
Target: Action: ReserveRoundtripFlights Action Input: {"airlines": " Delta Airlines", "departure_date": "2024-01-05", "destination_city": "Chicago", "origin_city": "New York", "return_date": "2024-01-11", "seating_class": "Economy", "number_stops": "1", "refundable": False} |
| Nonsensical Tool Names | The designation of the tool lacks significance, for instance, "abc". | System Prompt: ...The following is a list of APIs and their parameters that you can use: { "name": "eee", "description": "Book an appointment at a dentist for a given time and date", "required_parameters": ["dentist_name", "appointment_time", "appointment_date"]...
User: I would like to book an appointment with Yvonne Yang at Greenview Dental Care on the 11th at 11:45. Do they offer cosmetic services?
Target: Action: eee Action Input:
{"appointment_date": "2024-01-11", "appointment_time": "11:45", "dentist_name": "Yvonne Yang"} |
| Determination of Specific Parameter Values | Parameter values must conform to specific criteria. | Tool Definition: "name": "searchHotel", "description": "To search for hotels based on a set of criteria including rating, type, amenities, location, and price range.", "optional_parameters": [..."name": "priceRange", "choices" = ["cheap", "moderate", "expensive"]
User: Can you help me find a moderately priced hotel in the centre with parking and internet availability?
Target: Action: SearchHotel Action Input: "area": "centre", "parking": True, "internet": True, "priceRange": "moderate", "return_info": ["all"] |
| Inability to Call Tools | The list of tools does not contain any suitable options. | System Prompt: You are a ...The following is a list of APIs and their parameters that you can use...
Prompt: Can you find me a drama movie with subtitles directed by Claire Denis?
Target: Thought: The user wants to find a drama movie with subtitles directed by Claire Denis. However, there is no available API in the provided list that supports the searching for movies. Therefore, it's not possible to meet the user's request with the current tools. |
| Interaction among Multiple Tools | In a single turn of dialogue, more than four tools are utilized. | User: I am looking for the date of a concert in New York, followed by the weather forecast for that day. And I want to find a dentist, book an appointment at 10:00 on Febrary 2nd. And please add the dentist's phone to my contact list. Thank you!
Target Action: FindEvents Action Input: {"event_type": "concert", "city": "New York"} Action : GetWeather Action Input: {"city": "New York", "date": "FindEvents.date"} Action: BookAppointment Action Input: {"dentist_name": "FindProvider.dentist_name", "appointment_time": "10:00", "appointment_date": "2024-02-02"} Action: AddNewContact Action Input: {"contact_name": "FindProvider.dentist_name", "phone_number": "FindProvider.phone_number"} |
| Multi-Turn Parameter Inheritance | The parameters for the current turn must be extracted from the historical dialogue. | User: Find some music please.
Assistant: Action: LookupMusic
Observation: {"artist": "Ariana Grande", "song_name": "Be My Baby"}
Assistant: Ok, how about Be My Baby by Ariana Grande?
User: That sounds good.
Assistant: Should I play this now?
User: Yes. Play on kitchen speaker please.
Target: Action: PlayMedia Action Input: { "playback_device": "kitchen speaker", "song_name": "Be My Baby" } |

---

**System Prompt (S-S)**

Please reply to the user based on their input and historical conversation information. You can choose to call external tools to implement it. Here are the call requirements and information about available APIs.

1. Please provide your thought process in \"Thought\", including user intent analysis, whether to call APIs, and how to call APIs.

2. When a user's request can be satisfied by calling an API, please provide the required calling information in the following format: \nAction: The name of the API to be called.\nAction Input: The parameter information required to call the API, in Json format.

3. If the user's needs can be met without calling the API, then no API call action will be made.

4. If there is a lack of mandatory information that makes it impossible to call a specific API, then no API call action will be made.

5. The name and parameter name of the API must be consistent with the provided API information. The value of the parameter should be extracted from the context, and the information should not be fictional.

6. If none of the apis provided are available to meet the user's requirements, no Action is taken.

7. If you need to call the API, your output format should be:

Thought: xxxx\nAction: xxxx\nAction Input: xxxx

For example:

Thought: The user is looking for a one−way flight for three people from Las Vegas to Atlanta on a specific date, March 5th. Let's call the SearchOnewayFlight API to find the available flights.\nAction: SearchOnewayFlight\nAction Input: {\"origin_city\": \"Las Vegas\", \"destination_city\": \"Atlanta\", \"departure_date\": \"2024−03−05\", \"seating_class\": \"Economy\"}

If you don't need to call the API, your output format should be:

Thought: xxxx

The following is a list of APIs and their parameters that you can use:

[apis information]

---

**System Prompt (S-M)**

Please reply to the user based on their input. You can choose to call external APIs to implement it. Here are the call requirements and information about available APIs.
1. Please provide your thought process in \"Thought\", including user intent analysis, whether to call APIs, and how to call APIs.
2. When a user's request can be satisfied by calling APIs, please provide the required calling information in the following format: \nAction: The name of the APIs.\nAction Input: the parameter information required to call APIs, in Json format. For example, \nAction: \"api_name_A\"\nAction Input: {\"parameter_name_A.1\": \"parameter_value_A.1\", ...}\nAction: \"api_name_B\"\nAction Input: {\"parameter_name_B.1\": \"parameter_value_B.2\", ...}, ...
3. There may be an interaction relationship between APIs, where the parameter value returned by the previous API call needs to be used as the parameter value for the next API call. Please use \"previous_API_name.return_parameter_name\" as the parameter value for the new API call.
4. Multiple APIs may need to be called to meet the user's needs. Please pay attention to the order of APIs' call.
5. There may be an interaction relationship between APIs, where the parameter value returned by the previous API call needs to be used as the parameter value for the next API call. Please use \"previous_API_name.return_parameter_name\" as the parameter value for the new API call.
6. The name and parameter name of the API must be consistent with the provided API information. The value of the parameter should be extracted from the context, and the information should not be fictional.
7. If you need to call the API, your output format should be:
\nThought: xxxx\nAction: xxxx\nAction Input: xxxx\nAction: xxxx\nAction Input:xxxx\nAction: xxxx\nAction Input: xxxx...
If you don't need to call the API, your output format should be:
\nThought: xxxx
The following is a list of APIs and their parameters that you can use:
[apis information]

System Prompt (M-S)

Please reply to the user based on their input and historical conversation information. You can choose to call external tools to implement it. Here are the call requirements and information about available APIs.
1. Please provide your thought process in \"Thought\", including user intent analysis, whether to call APIs, and how to call APIs.
2. When a user's request can be satisfied by calling an API, please provide the required calling information in the following format: \nAction: The name of the API to be called.\nAction Input: The parameter information required to call the API, in Json format.
3. \"Observation\" is the information returned by API calls.
4. If the user's needs can be met without calling the API, then no API call action will be made.
5. If there is a lack of mandatory information that makes it impossible to call a specific API, then no API call action will be made.
6. Note that if the dialogue history already contains the required information, there is no need to call the tool again.
7. The name and parameter name of the API must be consistent with the provided API information. The value of the parameter should be extracted from the context, and the information should not be fictional.
8. If you need to call the API, your output format should be:
\nThought: xxxx\nAction: xxxx\nAction Input: xxxx
Therefore, if you don't need to call the API, your output format should be:
\nThought: xxxx
The following is a list of APIs and their parameters that you can use:
[apis information]

System Prompt (M-M)

Please reply to the user based on their input and history conversation. You can choose to call external APIs to implement it. Here are the call requirements and information about available APIs.

1. Please provide your thought process in \"Thought\", including user intent analysis, whether to call APIs, and how to call APIs.

2. When a user's request can be satisfied by calling APIs, please provide the required calling information in the following format: \nAction: The name of the APIs.\nAction Input: the parameter information required to call APIs, in Json format. For example, \nAction: \"api_name_A\"\nAction Input: {\"parameter_name_A.1\": \"parameter_value_A.1\", ...}\nAction: \"api_name_B\"\nAction Input: {\"parameter_name_B.1\": \"parameter_value_B.2\", ...}, ...

3. There may be an interaction relationship between APIs, where the parameter value returned by the previous API call needs to be used as the parameter value for the next API call. Please use \"previous_API_name.return_parameter_name\" as the parameter value for the new API call.

4. Multiple APIs may need to be called to meet the user's needs. Please pay attention to the order of APIs' call.

5. The name and parameter name of the API must be consistent with the provided API information. The value of the parameter should be extracted from the context, and the information should not be fictional.

6. If you need to call the API, your output format should be:

Thought: xxxx\nAction: xxxx\nAction Input: xxxx\nAction: xxxx\nAction Input:xxxx\nAction: xxxx\nAction Input: xxxx...

If you don't need to call the API, your output format should be:

Thought: xxxx

The following is a list of APIs and their parameters that you can use:

[apis information]

**Examples for computing metrics.** The examples are listed below:

---

**Tool Selection Accuracy (abbr. TS)**   (for S-S, M-S)

✓ **Positive Example:**

- **User**: "Search for the latest news on artificial intelligence."

- **Assistant**: OK! Let me search the web. `search_web(query="latest news on AI")`

- **Ground Truth Tool**: `search_web`

- **Assistant's Tool**: `search_web`

- **Score**: TS = 1

- **Explanation**: The assistant correctly selected the tool `search_web`, matching the ground truth.

✗ **Negative Example:**

- **User**: "Calculate the square root of 16."

- **Assistant**: OK! Let me use the calculator. `search_web(query="square root of 16")`

- **Ground Truth Tool**: `calculate`

- **Assistant's Tool**: `search_web`

- **Score**: TS = 0

- **Explanation**: The assistant incorrectly selected `search_web` instead of the correct tool `calculate`.

---

**Parameter Selection Accuracy (abbr. PS)**   (for S-S, M-S)

✓ **Positive Example:**

- **User**: "Translate 'good evening' to German."

- **Assistant**: Sure, translating now. `translate(text="good evening", target_language="German")`

- **Ground Truth Parameters**: {text="good evening", target_language="German"}

- **Assistant's Parameters**: {text="good evening", target_language="German"}

- **Score**: PS = 1

- **Explanation**: The assistant provided the correct parameters matching the ground truth.

✗ **Negative Example:**

- **User**: "Set an alarm for 6 AM tomorrow."

- **Assistant**: Setting your alarm. `set_alarm(time="6 PM", name="tomorrow")`

- **Ground Truth Parameters**: {time="6 AM", date="tomorrow"}

- **Assistant's Parameters**: {time="6 PM", name="tomorrow"}

- **Score**: PS = 0

- **Explanation**: The assistant incorrectly set the value for the time parameter. The assistant incorrectly set the key for the date parameter.

---

**Success Rate (abbr. SR)**   (for M-S, M-M)

✓ **Positive Example:**

- **User**: "I need to send an email to Alex."

- **Assistant**: Sure, composing an email. `compose_email(recipient="Alex", subject="Hello", body="Hi Alex, how are you?")`

- **User**: "Also, schedule a meeting with him next Monday at 2 PM."

- **Assistant**: Scheduling the meeting. `schedule_meeting(participant="Alex", date="next Monday", time="2 PM")`

- **Ground Truth**: No errors throughout all turns.

- **Score**: SR = 1

- **Explanation**: The assistant correctly used the tools in all dialogue turns without any errors.

✗ **Negative Example:**

- **User**: "Find the fastest route to the airport."

- **Assistant**: Finding the fastest route. `get_directions(destination="airport")`

- **User**: "Book a taxi to pick me up in 30 minutes."

- **Assistant**: Booking a taxi. `book_taxi(pickup_time="in 3 minutes")`

- **Ground Truth**: Assistant should have set the pickup time to "in 30 minutes".

- **Score**: SR = 0

- **Explanation**: The assistant made an error in the second turn (incorrect pickup time), so the entire dialogue is considered unsuccessful.

---

**Averaged Turn Success Rate (abbr. ATS)**   (for M-S, M-M)

✓ **Positive Example:**

- **Turn 1**:

- **User**: "What's on my schedule today?"

- **Assistant**: Checking your schedule. `get_schedule(date="today")`

- **Turn Success**: 1

- **Turn 2**:

- **User**: "Add a dentist appointment at 4 PM."

- **Assistant**: Adding the appointment. `add_event(event="dentist appointment", time="4 PM", date="today")`

- **Turn Success**: 1

- **Turn 3**:

- **User**: "Remind me an hour before."

- **Assistant**: Setting a reminder. `set_reminder(event="dentist appointment", time="3 PM", date="today")`

- **Turn Success**: 1

- **Score**: ATS = (1 + 1 + 1)/3 = 1.0

- **Explanation**: All turns were executed correctly.

✗**Negative Example:**

- **Turn 1**:

- **User**: "Play my workout playlist."

- **Assistant**: Playing your playlist. `play_music(playlist="workout")`

- **Turn Success**: 1

- **Turn 2**:

- **User**: "Skip to the next song."

- **Assistant**: Restarting the song. `play_music(action="restart")`

- **Turn Success**: 0 (Incorrect action parameter)

- **Turn 3**:

- **User**: "Increase the volume."

- **Assistant**: Increasing volume. `adjust_volume(level="up")`

- **Turn Success**: 1

- **Score**: ATS = (1 + 0 + 1)/3 = 0.6667

- **Explanation**: One out of three turns was unsuccessful, leading to an ATS of approximately 0.6667.

---

**Soft Averaged Turn Success Rate (abbr. SATS)**    (for M-S, M-M)

✓**Positive Example:**

- **Turn 1**:

- **User**: "What's the current time in Tokyo?"

- **Assistant**: The current time is... `get_time(location="Tokyo")` → `3:00 PM`

- **Turn Success**: 1

- **Turn 2**:

- **User**: "Schedule a call with the Japan office at that time tomorrow."

- **Assistant**: Scheduling the call. `schedule_call(participant="Japan office", time="3:00 PM", date="tomorrow")`

- **Turn Success**: 1

- **Score**: SATS = (1 + 1)/2 = 1.0

- **Explanation**: Both turns were successful, so the SATS is 1.

✗**Negative Example:**

- **Turn 1**:

- **User**: "Check the weather in New York."

- **Assistant**: Checking the weather. `get_weather(location="New York")`

- **Turn Success**: 1

- **Turn 2**:

- **User**: "Schedule a meeting tomorrow at 10 AM."

- **Assistant**: Scheduling the meeting. `schedule_meeting(date="tomorrow", time="10 AM")`

- **Turn Success**: 1

- **Turn 3**:

- **User**: "Set an alarm for 9 AM."

- **Assistant**: Setting an alarm. `set_alarm(time="9 PM")`

- **Turn Success**: 0 (Incorrect time parameter)

- **Turn 4**:

- **User**: "Remind me to call John at 11:00 AM."

- **Assistant**: Setting a reminder. `set_reminder(event="call John", time="11:00 AM")`

- **Turn Success**: 1

- **Turn 5**:

- **User**: "Send an email to Sarah."

- **Assistant**: Composing an email. `compose_email(recipient="Sarah", subject="Hello", body="Hi Sarah, ...")`

- **Turn Success**: 1

**Calculations**:

- Last incorrect turn index $i = 3$

- For $j = 1$ and $j = 2$:

- Since $j < i$, Score = 1

- For $j = 3$:

- Turn is incorrect, Score = 0

- For $j = 4$:

- $j > i$, Score = $1 - e^{-(4-3)} = 1 - e^{-1} \approx 0.6321$

- For $j = 5$:

- $j > i$, Score = $1 - e^{-(5-3)} = 1 - e^{-2} \approx 0.8647$

- **Score**: SATS = $\frac{1+1+0+0.6321+0.8647}{5} \approx \frac{3.4968}{5} \approx 0.6994$

- **Explanation**: The error in the third turn reduces the scores of subsequent turns due to the exponential decay, resulting in a SATS of approximately 0.6994.

---

**Task Process Rate (abbr. TPR)**    (for M-S, M-M)

✓**Positive Example:**

- **Total Turns**: $n = 4$

- **First Incorrect Turn Index**: No incorrect turns.

- **Score**: TPR = $\frac{n}{n} = 1$

- **Explanation**: Since there are no errors, the task was processed completely.

✗**Negative Example:**

- **Total Turns**: $n = 5$

- **First Incorrect Turn Index**: $i = 3$ (error occurs at turn 3)

- **Score**: TPR = $\frac{i-1}{n} = \frac{2}{5} = 0.4$

- **Explanation**: The task process rate indicates the proportion of the task completed before the first error.

---

**Tool Number Accuracy (abbr. TN)**   (for S-M, M-M)

✓**Positive Example:**

- **Ground Truth Tool List (GT)**: {search_web, summarize_text, translate_text}

- **Assistant's Predicted Tool List (Pred)**: {search_web, summarize_text, translate_text}

- **Score**: TN = $\frac{|\text{Pred} \cap \text{GT}|}{|\text{Pred} \cup \text{GT}|} = \frac{3}{3} = 1.0$

- **Explanation**: All tools predicted by the assistant match the ground truth.

✗**Negative Example:**

- **Ground Truth Tool List (GT)**: {search_web, translate_text}

- **Assistant's Predicted Tool List (Pred)**: {search_web, play_music, set_alarm}

- **Score**: TN = $\frac{|\text{Pred} \cap \text{GT}|}{|\text{Pred} \cup \text{GT}|} = \frac{1}{4} = 0.25$

- **Explanation**: Only search_web is common between the predicted and ground truth lists, out of four unique tools.

---

**Tool Order Accuracy (abbr. TO)**   (for S-M, M-M)

✓**Positive Example:**

- **Ground Truth Tool Sequence (GT)**: [search_web, extract_data, generate_report]

- **Assistant's Predicted Tool Sequence (Pred)**: [search_web, extract_data, generate_report]

- **Longest Common Subsequence (LCS)**: [search_web, extract_data, generate_report]

- **Starting Position (i)**: 1

- **Total Tools in Pred**: $|\text{Pred}| = 3$

- **Decay Coefficient (t)**:

$t = \cos\left(\frac{\pi}{2} \times \frac{i}{|\text{Pred}|}\right) = \cos\left(\frac{\pi}{2} \times \frac{1}{3}\right) \approx 0.8660$

- **Score**:

TO = $t \times \frac{|\text{LCS}|}{|\text{GT}|} = 0.8660 \times \frac{3}{3} = 0.8660 \times 1 = 0.8660$

- **Explanation**: The assistant's sequence perfectly matches the ground truth, resulting in a high TO score.

✗**Negative Example:**

- **Ground Truth Tool Sequence (GT)**: [get_weather, plan_route, book_hotel]

- **Assistant's Predicted Tool Sequence (Pred)**: [book_hotel, plan_route, get_weather]

- **Longest Common Subsequence (LCS)**: [plan_route]

- **Starting Position (i)**: 2

- **Total Tools in Pred**: $|\text{Pred}| = 3$

- **Decay Coefficient (t)**:

$t = \cos\left(\frac{\pi}{2} \times \frac{i}{|\text{Pred}|}\right) = \cos\left(\frac{\pi}{2} \times \frac{2}{3}\right) \approx 0.5000$

- **Score**:

$\text{TO} = t \times \frac{|\text{LCS}|}{|\text{GT}|} = 0.5000 \times \frac{1}{3} \approx 0.1667$

- **Explanation**: Only one tool matches in sequence, and it starts at the second position, leading to a low TO score.

Table 9: Effect of MTU-Instruct based on different models on MTU-Eval **(S-S & M-S)**. "S-S" and "M-S" denote "Single-Turn Single-Tool" and "Multi-Turn Single-Tool" settings, respectively. All the baselines are instruction-tuned models.

| Models | S-S | | | M-S | | | | | | |
|---|---|---|---|---|---|---|---|---|---|---|
| | TN | TO | Avg. | TN | TO | ATS | SATS | SR | TPR | Avg. |
| *Normal Set* | | | | | | | | | | |
| Qwen2.5-7B | 75.00 | 48.21 | 61.61 | 74.83 | 54.76 | 51.84 | 43.25 | 5.56 | 17.88 | 41.35 |
| MTU-Qwen2.5-7B | 96.15 | 44.23 | **70.19** | 81.63 | 67.01 | 66.47 | 57.49 | 9.26 | 29.56 | **51.90** |
| Qwen2.5-72B | 82.14 | 66.07 | 74.11 | 82.31 | 68.03 | 67.24 | 59.97 | 14.81 | 38.31 | 51.55 |
| MTU-Qwen2.5-72B | 96.15 | 55.77 | **75.96** | 91.16 | 76.19 | 72.22 | 65.88 | 20.37 | 44.25 | **61.68** |
| LLaMA3-70B | 86.54 | 57.69 | 72.12 | 79.25 | 61.90 | 62.18 | 54.81 | 14.81 | 32.33 | 50.88 |
| MTU-LLaMA3-70B | 91.38 | 53.16 | **72.27** | 89.12 | 71.09 | 68.69 | 60.73 | 12.96 | 37.28 | **56.64** |
| *Hard Set* | | | | | | | | | | |
| Qwen2.5-7B | 75.00 | 50.00 | 62.50 | 74.86 | 46.93 | 42.48 | 33.27 | 0.00 | 10.67 | 34.70 |
| MTU-Qwen2.5-7B | 75.96 | 51.36 | **63.66** | 83.24 | 66.48 | 63.95 | 54.12 | 0.00 | 26.78 | **49.09** |
| Qwen2.5-72B | 80.36 | 67.86 | 74.11 | 83.80 | 60.89 | 58.11 | 50.58 | 3.23 | 28.72 | 47.55 |
| MTU-Qwen2.5-72B | 83.44 | 69.71 | **76.58** | 92.18 | 72.63 | 69.24 | 61.50 | 6.45 | 34.05 | **56.01** |
| LLaMA3-70B | 82.14 | 60.71 | 71.43 | 75.42 | 52.51 | 51.68 | 42.03 | 3.23 | 17.53 | 40.40 |
| MTU-LLaMA3-70B | 89.79 | 66.79 | **78.29** | 94.41 | 76.54 | 72.06 | 65.10 | 16.13 | 41.37 | **60.94** |

Table 10: Effect of MTU-Instruct based on different models on MTU-Eval **(S-M & M-M)**. "S-M" and "M-M" denote "Single-Turn Multi-Tool" and "Multi-Turn Multi-Tool" settings, respectively. All the baselines are instruction-tuned models.

| Models | S-M | | | M-M | | | | | | |
|---|---|---|---|---|---|---|---|---|---|---|
| | TN | TO | Avg. | TN | TO | ATS | SATS | SR | TPR | Avg. |
| *Normal Set* | | | | | | | | | | |
| Qwen2.5-7B | 28.18 | 32.16 | 30.17 | 44.65 | 47.39 | 43.37 | 42.37 | 7.14 | 26.54 | 35.24 |
| MTU-Qwen2.5-7B | 32.89 | 33.10 | **33.00** | 49.14 | 50.04 | 46.51 | 40.45 | 7.14 | 29.17 | **37.08** |
| Qwen2.5-72B | 54.18 | 56.91 | 55.55 | 59.88 | 60.37 | 59.88 | 57.58 | 14.29 | 53.25 | 50.87 |
| MTU-Qwen2.5-72B | 57.45 | 61.66 | **59.56** | 63.88 | 67.96 | 64.39 | 61.34 | 19.77 | 59.09 | **56.07** |
| LLaMA3-70B | 26.85 | 32.68 | 29.76 | 33.60 | 35.71 | 26.94 | 23.82 | 0.00 | 17.86 | 22.99 |
| MTU-LLaMA3-70B | 56.42 | 59.73 | **58.08** | 63.33 | 64.28 | 62.50 | 60.20 | 19.05 | 55.75 | **54.15** |
| *Hard Set* | | | | | | | | | | |
| Qwen2.5-7B | 19.26 | 19.20 | 19.23 | 38.93 | 39.71 | 36.53 | 34.67 | 2.70 | 31.04 | 30.60 |
| MTU-Qwen2.5-7B | 40.11 | 36.77 | **38.44** | 39.62 | 40.21 | 38.96 | 33.41 | 5.30 | 33.51 | **31.84** |
| Qwen2.5-72B | 54.18 | 56.91 | 55.55 | 50.70 | 51.29 | 51.40 | 47.43 | 2.70 | 39.59 | 40.52 |
| MTU-Qwen2.5-72B | 57.45 | 57.45 | **57.45** | 55.79 | 56.95 | 52.97 | 47.17 | 5.41 | 35.05 | **42.22** |
| LLaMA3-70B | 19.70 | 21.64 | 20.67 | 31.74 | 33.13 | 24.86 | 20.63 | 0.00 | 13.02 | 20.56 |
| MTU-LLaMA3-70B | 41.83 | 39.85 | **40.84** | 50.00 | 50.56 | 48.69 | 46.62 | 10.81 | 43.06 | **41.62** |

# E  MORE EXPERIMENTAL ANALYSIS

**Training on other open-source models.**    To further validate the effectiveness of MTU-Instruct, we conduct additional experiments to evaluate the performance of several open-source models fine-tuned using MTU-Instruct on the MTU-Eval. Specifically, we select three models: Qwen2.5-7B, Qwen2.5-72B, and LLaMA3-70B. The detailed evaluation results are presented in Tables 10 and 12. The results indicate that the fine-tuned models demonstrate consistent performance improvements over the base model across all the settings.

**Detailed Evaluation Results on Hard Set.**    In Table 11 and Table 9, we present the performance of various LLMs evaluated on the hard subset of MTU-Eval. Our analysis indicates that among closed-source models, GLM-4-Plus excels in handling single-tool scenarios, surpassing even GPT-4.

Table 11: Results of different models on the **hard** set of MTU-Eval (**S-S & M-S**). "S-S" and "M-S" denote "Single-Turn Single-Tool" and "Multi-Turn Single-Tool" settings, respectively. All the baselines are instruction-tuned models.

| Models | S-S | | | M-S | | | | | | |
|---|---|---|---|---|---|---|---|---|---|---|
| | TS | PS | Avg. | TS | PS | ATS | SATS | SR | TPR | Avg. |
| *Closed-Source Large Language Models* | | | | | | | | | | |
| GPT-4 | 88.46 | 67.31 | 77.88 | **85.47** | 62.01 | 55.77 | 46.57 | 0.00 | 17.80 | 44.61 |
| GPT-3.5 | 50.00 | 33.93 | 41.96 | 71.51 | 43.02 | 39.45 | 29.33 | 0.00 | 1.83 | 30.86 |
| Qwen-Max | 88.46 | 67.31 | 77.88 | 80.45 | 56.98 | 50.54 | 41.77 | 3.23 | 19.67 | 42.11 |
| GLM-4-Plus | **92.31** | **73.08** | **82.69** | 84.92 | **63.69** | **58.40** | **48.79** | **6.45** | **23.44** | **47.61** |
| DeepSeek V2.5 | 90.38 | 71.15 | 80.77 | 83.24 | 58.10 | 56.10 | 47.28 | 3.23 | 21.66 | 44.94 |
| *Open-Source Large Language Models* | | | | | | | | | | |
| LLaMA2-7B | 32.14 | 25.00 | 28.57 | 34.64 | 26.82 | 22.32 | 16.32 | 0.00 | 2.70 | 17.13 |
| LLaMA2-70B | 32.14 | 25.00 | 28.57 | 52.51 | 30.17 | 28.45 | 21.62 | 0.00 | 8.03 | 23.46 |
| LLaMA3-8B | 33.93 | 17.86 | 25.89 | 37.99 | 13.97 | 11.64 | 8.15 | 0.00 | 5.38 | 12.85 |
| LLaMA3-70B | 82.14 | 60.71 | 71.43 | 75.42 | 52.51 | 51.68 | 42.03 | 3.23 | 17.53 | 40.40 |
| Qwen1.5-14B | 44.64 | 44.64 | 44.64 | 45.81 | 45.81 | 40.28 | 32.05 | 0.00 | 12.40 | 29.39 |
| Qwen1.5-72B | 57.69 | 55.77 | 56.73 | 48.11 | 48.11 | 40.10 | 31.62 | 0.00 | 11.57 | 29.92 |
| Qwen2-7B | 67.86 | 50.00 | 58.93 | 64.80 | 36.31 | 32.90 | 24.91 | 0.00 | 13.48 | 28.73 |
| Qwen2-72B | 78.93 | 57.86 | 68.40 | 78.77 | 50.84 | 47.19 | 37.20 | 0.00 | 16.50 | 38.42 |
| Mistral-7B | 36.54 | 17.31 | 26.92 | 63.69 | 34.64 | 31.29 | 23.90 | 0.00 | 2.74 | 26.04 |
| ChatGLM3-6B | 13.64 | 4.55 | 9.09 | 21.44 | 5.21 | 4.13 | 2.61 | 0.00 | 0.00 | 5.57 |
| GLM-4-9B | 59.62 | 34.62 | 47.12 | 64.80 | 37.99 | 36.56 | 29.42 | 0.00 | 12.52 | 30.22 |
| *Tool-Use-Specific Large Language Models* | | | | | | | | | | |
| ToolLLaMA2-7B | 32.69 | 3.85 | 18.27 | 25.70 | 3.35 | 3.50 | 2.60 | 0.00 | 0.65 | 10.19 |
| MTU-LLaMA(ours) | 51.79 | 28.57 | 40.18 | 78.21 | 59.22 | 57.35 | 46.90 | 0.00 | 16.93 | 43.10 |

Conversely, Qwen-Max demonstrates superior performance in multi-turn and multi-tool scenarios, also outperforming GPT-4. In single-turn multi-tool scenarios, GPT-4 and DeepSeek V2.5 exhibit relatively better performance. Furthermore, there remains a significant gap between open-source models and their closed-source counterparts. Among the open-source models, our MTU-LLaMA, Qwen2-72B, and LLaMA3-70B lead the performance metrics. Specifically, Qwen2-72B shows a slight advantage in multi-tool scenarios, while LLaMA3-70B excels in single-tool scenarios. MTU-LLaMA, however, demonstrates a balanced ability across almost all evaluated settings.

**Error Analysis.** The error cases of GPT-4 results on S-S, M-S, and S-M are enumerated in Tables 13, 14, and 15, respectively. Additionally, a comprehensive breakdown of error frequencies across different models and error categories is illustrated in Table 16.

**Details on the Human Evaluation.** Five annotators with extensive expertise in large language models (LLM) are engaged to assess the prediction outcomes. The annotators assign win/lose scores where a score of 1 represents a victory for GPT-3.5, a score of 0 indicates a tie, and a score of -1 signifies a win for LLaMA3-8B.

Table 12: Results of different models on the **hard** set of MTU-Eval **(S-M & M-M)**. "S-M" and "M-M" denote "Single-Turn Multi-Tool" and "Multi-Turn Multi-Tool" settings, respectively. All the baselines are instruction-tuned models.

| Models | S-M | | | M-M | | | | | | |
|---|---|---|---|---|---|---|---|---|---|---|
| | TN | TO | Avg. | TN | TO | ATS | SATS | SR | TPR | Avg. |
| *Closed-Source Large Language Models* | | | | | | | | | | |
| GPT-4 | **57.77** | **58.37** | **58.07** | **57.08** | 58.31 | 49.28 | 45.44 | 0.00 | 38.06 | 41.36 |
| GPT-3.5 | 17.75 | 19.04 | 18.39 | 21.31 | 22.39 | 15.09 | 10.41 | 0.00 | 2.03 | 11.87 |
| Qwen-Max | 23.75 | 24.28 | 24.01 | 57.04 | **58.56** | **49.96** | **49.13** | **8.11** | **47.70** | **45.08** |
| GLM-4-Plus | 32.43 | 29.36 | 30.90 | 52.74 | 53.74 | 47.03 | 43.11 | 5.41 | 35.18 | 39.53 |
| DeepSeek V2.5 | 40.06 | 39.96 | 40.01 | 43.51 | 45.21 | 35.45 | 32.49 | 0.00 | 27.07 | 30.62 |
| *Open-Source Large Language Models* | | | | | | | | | | |
| LLaMA2-7B | 2.14 | 2.56 | 2.35 | 15.14 | 15.27 | 15.00 | 13.69 | 0.00 | 11.44 | 11.76 |
| LLaMA2-70B | 1.39 | 2.08 | 1.74 | 20.23 | 20.74 | 20.54 | 20.04 | 0.00 | 19.19 | 16.79 |
| LLaMA3-8B | 9.25 | 10.57 | 9.91 | 11.45 | 12.60 | 6.26 | 4.33 | 0.00 | 0.68 | 5.89 |
| LLaMA3-70B | 19.70 | 21.64 | 20.67 | 31.74 | 33.13 | 24.86 | 20.63 | 0.00 | 13.02 | 20.56 |
| Qwen1.5-14B | 12.23 | 13.39 | 12.81 | 15.14 | 16.92 | 11.98 | 8.82 | 0.00 | 3.38 | 9.37 |
| Qwen1.5-72B | 19.27 | 18.42 | 18.85 | 26.92 | 27.07 | 21.71 | 18.61 | 0.00 | 13.29 | 17.93 |
| Qwen2-7B | 15.24 | 16.37 | 17.50 | 16.03 | 16.41 | 13.92 | 10.37 | 0.00 | 4.28 | 10.17 |
| Qwen2-72B | 35.12 | 36.13 | 37.14 | 35.05 | 36.83 | 30.72 | 27.52 | 0.00 | 20.68 | 25.13 |
| Mistral-7B | 10.18 | 12.79 | 11.48 | 17.05 | 17.81 | 13.38 | 10.69 | 0.00 | 6.08 | 10.84 |
| ChatGLM3-6B | 4.58 | 6.57 | 5.58 | 16.54 | 16.67 | 13.92 | 12.32 | 0.00 | 9.23 | 11.45 |
| GLM-4-9B | 20.00 | 17.96 | 18.98 | 17.43 | 17.94 | 12.75 | 8.31 | 0.00 | 0.68 | 9.52 |
| *Tool-Use-Specific Large Language Models* | | | | | | | | | | |
| ToolLLaMA2-7B | 0.51 | 0.51 | 0.51 | 3.82 | 3.82 | 3.15 | 2.32 | 0.00 | 0.90 | 2.34 |
| MTU-LLaMA(ours) | 39.77 | 38.85 | 39.31 | 35.90 | 36.67 | 34.14 | 27.37 | 0.00 | 14.14 | 24.70 |

Table 13: Error cases of GPT-4 results on S-S.

| Type | Description | Case |
|---|---|---|
| **Tool Selection** | The model incorrectly selects an unintended tool. | User: Please book 5 economy bus tickets from Sacramento to San Diego for the 10th of this month.
Predict: Action: FindBus Action Input: {"departure_date" : "2024-01-10", ...
Target: Action: BuyBusTicket Action Input: {"departure_date" : "2024-01-10", ... |
| **Paramete Omission** | The model omits necessary parameters in the results. | User: I need to find an affordable round trip flight from Washington to Mexico for 4 people, departing on the 10th and returning on the 13th of this month.
Predict: Action: SearchRoundtripFlights Action Input: {"origin_city" : "washington", "destination_city":"mexico", "departure_date": "2024-01-10", "return_date": "2024-01-13", "passengers" : "4"}
Target: Action: SearchRoundtripFlights Action Input: {"origin_city" : "washington", "destination_city":"mexico", "departure_date": "2024-01-10", "return_date": "2024-01-13", "passengers" : "4", "seating_class" : "economy" } |
| **Parameter Misalignment** | The model did not populate the parameters according to the specified criteria. | User: I need to find a round trip flight to NYC from London, leaving day after tomorrow and returning on the 9th, in Economy class.
Predict: Action: SearchRoundtripFlights Action Input: {"departure_date" : "2024-01-06", "return_date" : "2024-01-09", "seating_class" : "Economy", "origin_airport" : "London", "destination_airport" : "nyc" }
Target: Action: SearchRoundtripFlights Action Input: {"departure_date" : "2024-01-06", "return_date" : "2024-01-09", "seating_class" : "Economy", "origin_airport" : "London", "destination_airport" : "New York" } |

Table 14: Error cases of GPT-4 results on M-S.

| Type | Description | Case |
|------|-------------|------|
| **Repeated Call** | When the information already exists in the previous conversation history, the tool should not be called repeatedly. | User: Milton Country Park
Assistant: ...Action: SearchAttraction...
Observation: {"name": "Milton Country Park", "address": "milton county park, milton", "type": "park", "entrance_fee": "free", **"phone": "765-987554"**}
Assistant: It's a free park on the north side of town.
User: Could I get the phone number for that?
Predict: Thought: The user wants the phone about Milton Country Park, so I need to use the searchAttraction API to get the details.
Action: SearchAttraction Action Input: ...
Target: Thought: The phone number has be received in the history, so there is no need to call any tool. The phone number is 765987554. |
| **Parameter Hallucination** | The model incorrectly assumes a parameter without sufficient context. | ...(Dialogue history)...
User: ... I need a bus for two people that would be leaving Portland on the 5th.
...(Dialogue history)...
Assistant: ...Action: FindBus...
Observation: {"fare": "39", "from_location": "Portland", "from_station": "Portland Bus Station", "leaving_date": "2024-01-05", "leaving_time": "10:20", "to_location": "Vancouver", "to_station":"Pacific Central Station", "transfers": "0", "travelers": "2"}
User: I want to reserve the hotel room.
Assistant: ...You would like to check in on 5th?
User: I want to check in on the 8th.
Predict: Thought: The user has provided a check-in date, which, combined with information from past queries (hotel name and destination), enables the ReserveHotel API to reserve a hotel for the user. Action: ReserveHotel Action Input: {"hotel_name": "Atrium Inn Vancouver", "check_in_date":"2024-01-08", "number_of_days": "1", "destination": "Vancouver"}
Target: Thought: A specific check-in date has been provided, but the duration of the stay is still unknown. For how many days? |
| **Parameter Inheritance** | The model should inherit the information parameters from the conversation history and update the current state in a timely manner. | User: Is Dumbo playing today in Santa Rosa?
Assistant: ...Action: GetTimesForMovie...
User: Could you get me a taxi there? How long is that?
Observation: {"genre": "Fantasy", "location": "Santa Rosa", "movie_name": "Dumbo", "price": "9","show_date": "2024-01-04", "show_time": "22:00", "show_type": "3d", **"street_address": "409 Aviation Boulevard"**, "theater_name": "Airport Stadium 12"}
Predict: Thought: The user want to get a taxi to go to the movie theater.
Action: GetRide Action Input: {"destination": " theater ", "number_of_riders": "1"}
Target: Thought: The user want to get a taxi to go to the movie which is at 409 Aviation Boulevard. I can call GetRide.
Action: GetRide Action Input: {"destination": " 409 Aviation Boulevard ", "number_of_riders": "1"} |

Table 15: Error cases of GPT-4 results on S-M.

| Type | Description | Case |
|------|-------------|------|
| **Call Fewer Tools** | The model fails to call all the necessary tools. | User: I need to set up an appointment with a dentist in Los Angeles who offers cosmetic services on June 15th at 2:00 pm. Afterwards, set me a reminder two hours before the appointment named "Dentist Appointment".
Predict:
Action: FindProvider...
Action: SetAppointmentReminder...
Target:
Action: FindProvider...
Action: BookAppointment ...
Action: SetAppointmentReminder... |
| **Call More Tools** | The model calls additional unnecessary tools. | User: Hi, can you help me find a song of 2019. And modify it as my 3pm alarm's sound?
Predict:
Action: LookupSong...
Action: GetAlarm ...
Action: ChangeAlarmSound...
Target:
Action: LookupSong...
Action: ChangeAlarmSound... |
| **Call Wrong Tool** | The model selects an inappropriate tool for the task, potentially leading to incorrect results. | User: I'm looking for a pop music to play and set as my alarm sound for my 3pm alarm today. Can you help me with that?
Predict:
Action: LookupSong...
Action: SetAlarm ...
Target:
Action: LookupSong...
Action: ChangeAlarmSound |

Table 16: Error Frequencies across Various Models and Error Types. **Bold** number signifies the most critical error type for each model. Underlined number represents the most critical model for each error type.

| Models | Action Error | | | | Parameter Error | Format Error | SUM |
|---|---|---|---|---|---|---|---|
| | **Missed** | **Excessive** | **Incorrect** | **SUM** | | | |
| *S-S* | | | | | | | |
| GPT4 | 0 | 10 | 3 | 13 | **26** | 4 | 43 |
| LLaMA3-70B-Instruct | 2 | 9 | 5 | 16 | **27** | 1 | 44 |
| ChatGLM4-9B | 16 | 4 | 8 | 28 | **35** | 1 | 64 |
| LLaMA3-8B-Instruct | 9 | 3 | 10 | 22 | **38** | 12 | 72 |
| MTU-LLaMA(Ours) | 8 | 6 | 10 | 24 | **26** | 0 | 50 |
| *S-M* | | | | | | | |
| GPT4 | 8 | 6 | 4 | 18 | **36** | 0 | 54 |
| LLaMA3-70B-Instruct | 9 | 9 | 13 | 31 | **56** | 4 | 91 |
| ChatGLM4-9B | 28 | 2 | 10 | 40 | **52** | 1 | 93 |
| LLaMA3-8B-Instruct | 8 | 25 | 11 | **44** | 22 | 28 | 94 |
| MTU-LLaMA(Ours) | 8 | 4 | 5 | 17 | **49** | 0 | 66 |
| *M-S* | | | | | | | |
| GPT4 | 25 | 18 | 4 | 47 | **96** | 14 | 157 |
| LLaMA3-70B-Instruct | 69 | 22 | 3 | **94** | 92 | 11 | 197 |
| ChatGLM4-9B | 111 | 9 | 9 | **129** | 112 | 40 | 281 |
| LLaMA3-8B-Instruct | 91 | 24 | 17 | **132** | 113 | 101 | 346 |
| MTU-LLaMA(Ours) | 56 | 24 | 3 | **83** | 73 | 30 | 186 |
| *M-M* | | | | | | | |
| GPT4 | 11 | 38 | 4 | **53** | 44 | 0 | 97 |
| LLaMA3-70B-Instruct | 8 | 114 | 7 | **129** | 62 | 5 | 196 |
| ChatGLM4-9B | 28 | 145 | 10 | **183** | 38 | 15 | 236 |
| LLaMA3-8B-Instruct | 2 | 207 | 4 | **213** | 6 | 46 | 265 |
| MTU-LLaMA(Ours) | 14 | 103 | 3 | **120** | 28 | 1 | 149 |
| **SUM** | **511** | **782** | **143** | **1436** | **1031** | **314** | **2781** |

