# OpenReview forum: "MTU-Bench: A Multi-granularity Tool-Use Benchmark for Large Language Models"
_ICLR.cc/2025/Conference — ICLR 2025 Poster_

### Official Review · Reviewer_p9mu · 2024-10-30

**Soundness:** 2
**Presentation:** 3
**Contribution:** 3
**Rating:** 6
**Confidence:** 5

**Summary:**

This paper proposes MTU-Bench, a benchmark dataset for evaluating the ability of a large language model(LLM) to invoke tools in multiple scenarios. MTU-Bench provides a more granular and detailed approach compared to previous studies in this domain by considering two key dimensions: (1) the number of tools that can be invoked within a conversation and (2) the number of rounds of tool call involved in multi-turn dialogues. Moreover, the construction pipeline for the MTU-Bench dataset is novel and carefully designed. It begins by collecting tasks from traditional conversation datasets, and these datasets are then transformed through a synthesis process into tool-oriented conversations, simulating realistic tool usage. This innovative approach serves as a scalable paradigm for expanding both the variety of tools and the diversity of conversations available for research. In addition, the paper proposes MTU-Eval, an automated evaluation framework that does not require an external LLM, which reduces the cost of evaluation to a large extent. Finally, the MTU-Instruct dataset is introduced for fine-tuning the tool-usage capabilities of the model, demonstrating the excellent performance of the fine-tuned model in a variety of complex tasks.

**Strengths:**

1. Originality: MTU-Bench introduces a multi-granularity evaluation framework, considering the number of tools invoked and the number of turns required for interaction. This multi-granularity framework captures a variety of real-world interactions that are often overlooked in existing benchmarks. The construction of this benchmark employs a unique synthesis pipeline by transforming task-oriented dialogue datasets into tool-usage datasets.

2. Quality: The paper evaluates multiple LLMs (both open-source and closed-source) across a range of scenarios, including multi-tool, multi-turn, and out-of-distribution (OOD) tasks, ensuring robustness and completeness in the experimental validation.

3. Clarity: The paper follows a clear narrative, starting with motivation and problem formulation, followed by methodology, experiments, and detailed discussions on results, which makes it easy for readers to follow the flow of ideas.

4. Significance: The paradigm of transforming existing dialogue datasets into tool-use datasets opens new possibilities for exploring the tool-use ability of LLMs. The automated evaluation framework MTU-Eval significantly reduces the cost and complexity of benchmarking LLMs

**Weaknesses:**

1. Unable to cover complex tool-calling scenarios: The idea of the paper is to convert the dialogue into a function call dataset, which is an interesting and extendable idea. However, this also introduces a bottleneck that the dataset may not cover too much complex queries that request the agent to perform more than 5 steps to finish (the paper also states that only an average of 2 calls per turn in the dataset).

2. Limitations of Virtual APIs: MTU-Bench generates all tools based on traditional conversation data, resulting in all APIs being fictitious. This approach may make LLM behave severely differently from when it calls real-world API, for example, if the model generates a wrong call in an executable env, it has the possibility to recover from its mistakes. (However, the reviewer does not expect this problem to be solved in this setting, but it would be good if the author could propose some nice practical thoughts to that)

3. Insufficiently Designed Evaluation Metrics: SATS and TPR overly emphasize the position of the first error, failing to adequately consider the LLM's ability to handle early mistakes. For example, if an LLM can disregard an early error and correctly execute subsequent tasks, it demonstrates robustness that the current metric does not capture.

**Questions:**

1.How to calculate the Parameter Selection Accuracy in single-turn scenarios? Specifically, is it determined solely by whether the LLM's output matches the ground truth exactly? There are instances where the parameters provided by the LLM might convey the same meaning as the ground truth but differ in specific characters or formatting. Would a more nuanced approach to assessing parameter selection accuracy be beneficial? Do you use the same way of defining correctness in multi-turn scenarios?

2. Since the tools called in MTU-Bench are all synthetic and do not use the real-world API, do you have any experiments that show that the metrics used in the paper are consistent across models when applying datasets built from real-world API calls?

3. In tool creation, how do you make sure that the tools you create make sense? Besides, since the tools are merged in the latter stage, how can you guarantee that the tool provided can fit the request by the current turn?

4. Most API doc are generated from GPT-4, which indicates that the distribution of API docs possibly falls into the distribution into the GPT4 capability distribution, how to overcome such bias?

5. Limited discussion of current tool-calling benchmarks
[1] T-Eval: Evaluating the Tool Utilization Capability of Large Language Models Step by Step, ACL 2024
[2] GTA: A Benchmark for General Tool Agents, NeurIPS 2024

6. Considering that this is a benchmark paper, the authors are highly suggested to provide the dataset data in the supple files, which allows the reviewer to have a glance at the quantity of the dataset in person, which is different from the dedicated demos selected in the supplementary.

7. The author shows the performance gain on the MTU-Eval, however, it is actually an in-domain SFT. What is the performance gain on another tool benchmark?

---

> ### Author Response · Authors · 2024-11-25
> **Responses for Q1-Q3**
>
> Thanks for your valuable review. Here are our explanations.
>
> Q1: Unable to cover complex tool-calling scenarios.
>
> A1: We appreciate your insights regarding complex tool-calling tasks. Actually, for our dataset, approximately **22% of the single dialogue involve more than two function calls**. It indicates that while the average is 2 calls per turn, a significant portion of the dataset covers more complex scenarios. We believe this distribution reflects real-world usage patterns, where many interactions are relatively simple but some require multi-step processes.
>
> In addition, to test the model's ability to handle complex tasks, we manually created some challenging task samples by merging single-tool examples. Approximately **22%** of the S-M samples require **five or more tool calls**, and this part of the data constitutes the hard set of S-M. The experimental results show that MTU-LLaMA improved the score on S-M hard set by **29%**. It demonstrates the great potential of our MTU-LLaMA in handling complex tasks.
>
> Q2: Limitations of virtual APIs and the possibility to recover from the mistakes.
>
> A2: The data in MTU-Bench is indeed not from real-world APIs and is non-executable. We acknowledge that it will affect research on the subsequent behavior of LLMs. However, **the focus of this paper is to propose an efficient data synthesis** strategy **to enhance the tool-use capability of the models.** Our extensive experimental results also indicate that LLM models can benefit from MTU-Bench. We have significantly improved the first-pass accuracy of LLMs when calling tools. Based on your question, we conduct experiments with simulated feedback to explore whether the model has the ability to recover from mistakes when evaluated using MTU-Eval.
>
> We try providing three types of feedback: a) tool parsing errors, b) incorrect tool selection, and c) incorrect parameter selection. We add the model's response and tools' feedback information to the historical conversation and prompt the model to generate again. Below are the results of our experiments with LLaMA3-8B-Instruct and MTU-LLaMA3-8b on the S-S setting.
>
> |  | LLaMA3-8b-Instruct | MTU-LLaMA3-8b |
> | --- | --- | --- |
> | Withour Feedback | 39.43 | 51.92 |
> | With Feedback | 49.04 (+ 9.61) | 55.77 (+ 3.85) |
>
> From the results, it can be seen that the **models have a certain ability to correct errors when receiving feedback from the tool.**  Although the focus of this paper is not on this aspect, we recognize that the ability to reflect is also crucial for tool-use ability. Therefore, we encourage researchers to conduct more rich and interesting research using our dataset MTU-Bench. We will also further explore the relationship between reflection ability and tool-use ability in future research. Thank you for your insight!
>
> Q3: Insufficient designed evaluation metrics.
>
> A3: We understand that you are concerned about the sufficiency of our metrics. However, it seems there is a slight misunderstanding. TPR captures how early in the dialogue the first mistake occurs, but **SATS focuses on the nearest error point to the current turn**. The closer the error is to the current turn, the greater its impact on the response. We designed the influence factor 1-e^(j-i) to reflect this impact.
>
> We acknowledge that LLMs have the ability to ignore errors and continue subsequent tasks, which is not what we intended to evaluate using TPR and SATS. Instead, **ATS addresses this capability, as it calculates the proportion of turns that are successfully completed on average.** ATS disregards the impact of early errors and evaluates the model's overall performance throughout the dialogue. Therefore, we believe that our metrics are sufficient to comprehensively assess the performance.

---

> ### Author Response · Authors · 2024-11-25
> **Responses for Q4-Q5**
>
> Q4: The influence of semantic ambiguity.
>
> A4: We indeed determine the Parameter Selection Accuracy by whether the model's output fully matches the ground truth. We acknowledge that there are situations where LLMs convey the same meaning as the ground truth but differ in specific characters or formatting. We encountered this issue during our research. When calculating whether the predicted answer matches the ground truth, we implemented **some fuzzy matching,** such as converting numbers (1 to "one," 2 to "two") and weeks to dates. We found that it **addressed the majority of discrepancies** in expressions that could lead to evaluation errors.
>
> Additionally, to ensure that the LLM returns parameter values in a standardized format, we provide the **detailed format requirements for parameters with strict formatting requirements** in the API documentation. For instance, for parameters related to time, we require the value to be returned in the 'HH:MM' format. This helps avoid calling errors caused by ambiguous parameter value requirements, making our evaluations as accurate as possible.
>
> In multi-turn scenarios, we apply the same method.
>
> Q5: Not real-world APIs.
>
> A5: We understand your concern about the validity of our metrics in real-world APIs. We verify our metrics on the **ToolTALK, which involves real-world APIs.** We select seven models to evaluate ToolTALK and calculate the scores for each model using both the evaluation metrics from the ToolTALK and the metrics we proposed. We aim to **compare the ranking of scores from the two sets of metrics to validate their consistency.** Our results are as follows:
>
> 1. The scores calculated by calling real API. (The metrics from ToolTALK)
>
> | **Model** | **precision** | **recall** | **action_pre** | **success_rate** | **Avg** |
> | --- | --- | --- | --- | --- | --- |
> | GPT-4 | 74.14 | 81.33 | 75.27 | 44.44 | **68.80** |
> | GPT-4o | 60.80 | 37.34 | 91.36 | 3.70 | **48.30** |
> | GPT-3.5 | 47.83 | 32.71 | 86.96 | 0.00 | **41.88** |
> | MTU-LLaMA-8B | 36.42 | 15.60 | 59.26 | 0.00 | **27.82** |
> | Qwen2.5-7B-Instruct | 35.03 | 21.56 | 48.61 | 0.00 | **26.30** |
> | Qwen2-7B-Instruct | 34.51 | 22.07 | 43.86 | 0.00 | **25.11** |
> 2. The scores calculated by our metrics.
>
> | **Model** | **TS** | **PS** | **ATS** | **SATS** | **SR** | **TPR** | **Avg** |
> | --- | --- | --- | --- | --- | --- | --- | --- |
> | GPT-4 | 51.08 | 51.60 | 44.90 | 39.74 | 6.90 | 27.13 | **36.89** |
> | GPT-4o | 40.88 | 41.00 | 40.27 | 35.98 | 7.14 | 26.60 | **31.98** |
> | GPT-3.5 | 33.80 | 34.14 | 33.14 | 30.17 | 6.90 | 23.69 | **26.97** |
> | MTU-LLaMA-8B | 30.25 | 31.23 | 30.87 | 28.51 | 3.45 | 22.82 | **24.52** |
> | Qwen2.5-7b-Instruct | 29.71 | 29.76 | 29.22 | 27.98 | 2.34 | 18.13 | **22.86** |
> | Qwen2-7b-Instruct | 22.99 | 23.01 | 23.55 | 20.77 | 0.00 | 16.01 | **17.72** |
>
> From the results, it can be seen that the order of the two sets of metrics is completely consistent: GPT-4 > GPT-4o > GPT-3.5 > MTU-LLaMA-8B > Qwen2.5-7B-Instruct > Qwen2-7B-Instruct. And we calculate the **Pearson correlation coefficient**, and the correlation between the average scores from the two sets of evaluation metrics reached **as high as 0.95**. These can reflect the effectiveness of our metrics on the evaluation set of real-world tool calls.

---

> ### Author Response · Authors · 2024-11-25
> **Responses for Q6-Q10**
>
> Q6: The validity of tools.
>
> A6: To make sure that the tools we create make sense, we set a lower threshold for initial clustering, with each cluster retaining one tool. After that, we manually eliminated duplicate tools. By manual verification, we can ensure that the final clustered tool names make sense.
>
> Q7: API docs fall into the GPT-4 distribution.
>
> A7: Actually, the distribution of synthesized APIs is very similar to that of real-world APIs: We compute the BGE-M3[1] embeddings of the tools in MTU-Bench and the real-world APIs[2]. The Wasserstein distance between the distributions of MTU tools and real-world tools is only 0.0063, demonstrating their highly consistent nature.
>
> [1]BGE M3-Embedding: Multi-Lingual, Multi-Functionality, Multi-Granularity Text Embeddings Through Self-Knowledge Distillation
>
> [2]ToolLLM: Facilitating Large Language Models to Master 16000+ Real-world APIs
>
> Besides, Although we used GPT-4 generate tool documentation, we conducted **a thorough manual review** of the final tool documentation **for all 136 tools** to ensure that all API documentation is accurate. We chose GPT-4 because it is currently the most powerful LLM, and much of the synthesized data from other LLMs is also constructed using GPT-4. Such work may indeed introduce some bias, but it is not the focus of this study, so we will include it in the limitations section for discussion.
>
> Q8: Limited discussion of current tool-use benchmarks.
>
> A8: We list the differences between these two works and MTU-Bench in the table below. We would like to share a few points:
>
> The motivation of T-Eval lies in the desire to evaluate the tool use capabilities of LLM in a more nuanced way by breaking down these capabilities into multiple subprocesses. It is different from our MTU-Bench. MTU-Bench introduces a multi-granularity tool usage benchmark to **comprehensively assess** the capabilities of LLM in **real-world tool usage scenarios**. While GAT also emphasizes the construction of tool invocation data from realistic scenarios, MTU-Bench shows clear advantages in **data production efficiency**, **coverage of complex scenarios**, **breadth of domain coverage**, and the **number of tools**. Additionally, MTU-Bench includes more than just a test set; we have also produced a **tool use trainset with a scale of 5w** data aimed at enhancing the LLM's tool use capabilities. Below are the detailed comparison results:
>
> |  | #Dialogues | #Tools | #Turn-#Tool | RealWorld | Auto. Eval | Eval. Range | Train | Test | OOD |
> | --- | --- | --- | --- | --- | --- | --- | --- | --- | --- |
> | T-Eval | 23,305 | 15 | S-S, S-M | ✗ | ✔️ | ①②③ | ✗ | ✔️ | ✗ |
> | GTA | 229 | 14 | S-S, S-M | ✔️ | ✔️ | ①②③ | ✗ | ✔️ | ✗ |
> | MTU-Bench（Ours） | 54,798 | 136 | S-S, S-M, M-S, M-M | ✔️ | ✔️ | ①②③④⑤⑥ | ✔️ | ✔️ | ✔️ |
>
> We have added the comparation to Table 1 of the paper. Thank you very much for your reminder!
>
> Q9: Data samples.
>
> A9: We will provide detailed information about the dataset and attach the dataset files in the **supplemental materials** so that reviewers can personally examine the quantity of the data.
>
> Q10: Generality performance.
>
> A10: We understand your concern about the generalizability of our dataset. Indeed, we have already considered it. To evaluate the generality of MTU-LLaMA, we measure its performance on the OOD test split of MTU-Bench and two other OOD tool-use benchmarks, i.e., API-Bank and ToolTalk. The detailed information is shown in Section 3.2.
>
> Thanks again for your valuable advice.

---

> ### Author Response · Authors · 2024-11-28
>
> Hi, Reviewer p9mu,
>
> Thanks for your advice. We believe we have addressed your concerns carefully. As the discussion is short, please let us know if you have other questions or comments.

---

### Official Review · Reviewer_VABn · 2024-11-01

**Soundness:** 3
**Presentation:** 3
**Contribution:** 2
**Rating:** 5
**Confidence:** 4

**Summary:**

This paper introduces MTU-Bench, a benchmark designed to evaluate large language models (LLMs) in terms of their ability to use tools across various scenarios. MTU-Bench addresses limitations in existing benchmarks by providing more comprehensive scenario coverage and reducing evaluation costs. It includes five distinct tool usage scenarios and relies on prediction results and ground truth for evaluation. The paper's key contributions include the MTU-Bench benchmark, the MTU-Instruct and MTU-Eval datasets, and the MTU-LLaMA model, which is fine-tuned to demonstrate strong tool-use capabilities. The experimental results highlight the benchmark's effectiveness in enhancing LLMs' tool-use skills.

**Strengths:**

Comprehensive Evaluation: The paper provides a thorough evaluation of both open-source and proprietary models using the newly proposed benchmark, covering a wide range of scenarios.

Detailed Experimental Setup: The experiments are well-designed and extensive, allowing for a clear comparison of LLMs' tool-use capabilities.

Improved Scenario Coverage: By incorporating multiple scenarios, including multi-turn and multi-tool settings, the benchmark offers a more nuanced evaluation of LLMs, which is a step forward from existing benchmarks.

Clarity and Structure: The paper is well-structured, making it easy to follow the methodology and understand the results.

**Weaknesses:**

Limited Novelty: While the benchmark offers more scenarios and finer-grained evaluations, it lacks a significant innovation or breakthrough that fundamentally advances the field. The paper needs to clearly articulate how these additions lead to new insights or directions in tool-use capabilities for LLMs.

Lack of Real-World Impact: The paper does not provide concrete examples or case studies demonstrating how the new benchmark can lead to improvements in real-world applications. For example, the introduction of the COCO dataset in the object detection field highlighted specific challenges that state-of-the-art methods at the time struggled with, such as detecting small objects, handling occlusions, and recognizing a wider variety of categories. This enabled researchers to evaluate and improve their models effectively. In contrast, MTU-Bench does not clearly show how it highlights current limitations of LLM tool-use or how it can similarly drive innovation in practical applications.

**Questions:**

The paper notes that existing benchmarks lack sufficient evaluation scenarios. Your proposed benchmark seems to merely add a few extra scenarios—how does this impact tool usage? Are there fundamental differences between the new scenarios and the previous ones? Without strong experimental evidence, it may appear that you are simply expanding the dataset. If your goal is to build a comprehensive evaluation suite, it seems to lack a thorough analysis and discussion of its completeness.

Your paper claims that previous datasets were not based on real-world data, yet the dataset you present is constructed using GPT-4 and existing datasets, rather than data collected from actual application users. How do these data fundamentally differ from previous datasets in accurately reflecting real-world scenarios?

Although your dataset is more detailed and extensive than previous ones, it remains unclear which specific challenges it addresses. Could combining existing evaluation datasets achieve similar results? What unique value does your benchmark provide that makes it indispensable for evaluating tool usage?

---

> ### Author Response · Authors · 2024-11-25
> **Responses for Q1-Q3**
>
> Thanks for your valuable advice. Here are some clarifications.
>
> Q1: Limited novelty.
>
> A1: The novelty of our benchmark is threefold. First, it features several key elements absent in previous works, as listed in Table 1. These include an in-the-wild query distribution, evaluation of tool number and order, and consideration of hard cases such as information missing or specific API calls. These features ensure our benchmark's diversity and granularity while aligning more closely with real-world use cases—an undoubtedly crucial aspect.
> Second, our work yields key experimental findings. For instance, we uncover a discrepancy in error turn positions between closed-source and open-source LLMs (line 398), and demonstrate the ineffectiveness of models tuned on synthetic instructions when applied to real-world scenarios.
> Third, these findings inspire future improvements. Our work verifies a more effective workflow for tool-use data synthesis: generating APIs from real-world user queries proves more effective than generating queries from existing APIs. Furthermore, we encourage future research to focus on improving tool-use robustness across longer turns, multiple tool calls, and challenging scenarios. Previous benchmarks, with their idealized simplicity, fail to capture the complexity of real-world use cases, leading to fast performance saturation. For example, we also add the experimental results of the GPT-o1 model, which also demonstrates limited performance.
>
> |  | Model | S-S | M-S | S-M | M-M |
> | --- | --- | --- | --- | --- | --- |
> | normal-set | gpt4 | 73.96 | 63.10 | 68.68 | 61.80 |
> | normal-set | o1-mini | 62.50 | 60.92 | 69.09 | 65.91 |
> | hard-set | gpt4 | 77.88 | 44.61 | 58.07 | 41.32 |
> | hard-set | o1-mini | 78.85 | 49.03 | 56.00 | 58.03 |
>
> Q2: Lack of real-world impact.
>
> A2: The scenario breakdown is a core motivation of our work, as illustrated in Table 1. We account for diverse real-world user queries in various scenarios, including single-turn single-tool, multi-turn multi-tool, and several hard cases as listed in Table 8. Furthermore, we highlight the current limitations of LLM tool-use in Figure 7 and Tables 13, 14, 15, 16. While we broadly categorize error patterns into format, action, and parameter errors in the main text, a detailed breakdown is provided in Appendix E.
> In the single-turn single-tool setting, error patterns include tool selection errors, parameter omissions, and parameter misalignments (Table 13). The multi-turn single-tool setting reveals cases such as repeated calls, parameter hallucinations, and parameter inheritance. In the single-turn multi-tool setting, errors manifest as calling fewer tools, more tools, or wrong tools.
> These error patterns offer valuable insights into current state-of-the-art LLMs and suggest directions for future work. To improve performance in multi-turn multi-tool (M-M) settings, we need to reduce tool hallucination rates. Enhancing models' instruction-following capabilities for formatting will mitigate format errors and improving long-context understanding will boost M-M performance. We can explore incorporating additional tool retrievers or providing more context for similar tools to address action errors. For parameter errors stemming from omissions, hallucinations, ambiguities, or complex reasoning, we encourage the community to focus on enhancing models' multi-hop reasoning and retrieval capabilities.
>
> Q3: Lack of a thorough analysis and discussion of its completeness.
>
> A3: As shown in Table 1, our benchmark not only introduces critical scenarios such as multi-turn and multi-tool settings, hard and OOD cases, and a broader evaluation range, but also bridges the gap between academic tool-use benchmarks and real-world use cases by leveraging real-world user instructions. This "wildness" is one of the fundamental differences between our benchmark and previous ones. The comprehensiveness of our benchmark is also enhanced by the real-world user query sampling, which covers a diverse range of tool-use cases including S-S, S-M, M-S, M-M, and the hard cases discussed in Table 8. These features ensure the diversity and granularity of our proposed benchmark while also yielding novel experimental findings, as presented in A1 and A2. Notably, we observe that as the number of dialogue turns or tools increases, the models' tool selection accuracy decreases (Figure 5). This finding suggests that stronger long-context understanding and multi-turn instruction-following abilities are crucial for LLMs to handle complex tool-use scenarios effectively. This underscores the importance of emphasizing multi-tool and multi-turn scenarios in our benchmark.

---

> ### Author Response · Authors · 2024-11-25
> **Responses for Q4**
>
> Q4: How does our data reflect real-world scenarios?
>
> A4: We collect real-world dialogues from some existing datasets originally not targetted at LLM tool-use: MultiWOZ[1], SGD[2], TaskMaster[3], MetaLWOZ[4], ATIS[5], and SNIPS[6]. These datasets are composed of real-world user dialogues. However, the previous works such as ToolBench[7] instead leverages synthesized dialogues with existing APIs. In contrast, our work real-world dialogues, while the GPT is merely used for reformatting the dialogues into tool-use versions. Below demonstrates an example showing how the real-world dialogues are formatted into tool-use formats without changing the content:
>
> ```
> User: Is there anywhere exciting to visit?
>
> Assistant: Thought: The user has not specified where they want to visit. I can not perform a useful task.
>
> Assistant: In which city?
>
> User: In san diego.
>
> Assistant: Thought: Now that I know the location, I can find attractions in San Diego.\nAction: FindAttractions \nAction Input: {'location': 'San Diego'}
>
> Observation: {'attraction_name': 'Balboa Park', 'category': 'Park', 'free_entry': 'False', 'good_for_kids': 'True', 'location': 'San Diego', 'phone_number': '619-239-0512'
>
> Assistant: There are 10 attractions that might interest you. You can visit a park called balboa park.
>
> User: Is there an entrance fee and what is their number?
>
> Assistant: Thought: The user is requesting for information that was already returned in the previous API call.
>
> Assistant: There is no entrance fee and the number is 619-239-0512.
>
> User: I would like go to a rock concert next Thursday.
>
> Assistant: Thought: I do not have enough information to provide an event. I need to know what type of event the user is interested in.
>
> Assistant: What do you find interesting?
>
> User: I want to hear music.
>
> Assistant: Thought: The user specified they are interested in music. I will find events that are categorized as \"Rock\" occurring in San Diego on the provided date.\nAction: FindEvents \nAction Input: {'category': 'Rock', 'city': 'San Diego', 'date': '2019-03-07', 'event_type': 'Music'}
>
> Observation: {'category': 'Rock', 'city': 'San Diego', 'date': '2019-03-07', 'event_name': 'Fidlar', 'event_type': 'Music', 'time': '17:30', 'venue': 'The Observatory North Park', 'venue_address': '2891 University Avenue'}
>
> Assistant: Fidlar is playing at the observatory north park.
> ```
>
> We claim that synthesizing dialogues based on real-world APIs is sub-optimal compared with our workflow, i.e., synthesizing APIs based on real-world dialogues. This is proved by our experimental results (ToolLLaMA v.s. MTU-LLaMA in Table 2, 3, 4, OOD performance of MTU-LLaMA in Table 5). This is intuitive since the synthesized dialogues are not aligned with real-world user needs. In contrast, we can find that the distribution of synthesized APIs is very similar to that of real-world APIs: We compute the BGE-M3[8] embeddings of the tools in MTU-Bench and the real-world APIs[7]. The Wasserstein distance between the distributions of MTU tools and real-world tools is only 0.0063, demonstrating their highly consistent nature.
>
> [1] MultiWOZ: a large-scale multi-domain wizard-of-oz dataset for task-oriented dialogue modelling
>
> [2]SGD: Towards scalable multi-domain conversational agents: The schema-guided dialogue dataset
>
> [3]Taskmaster-1: Toward a realistic and diverse dialog dataset
>
> [4]MetaLWOZ: Fast domain adaptation for goal-oriented dialogue using a hybrid generative-retrieval transformer
>
> [5]The ATIS spoken language systems pilot corpus
>
> [6]Unsupervised transfer learning for spoken language understanding in intelligent agents
>
> [7]ToolLLM: Facilitating Large Language Models to Master 16000+ Real-world APIs
>
> [8]BGE M3-Embedding: Multi-Lingual, Multi-Functionality, Multi-Granularity Text Embeddings Through Self-Knowledge Distillation

---

> ### Author Response · Authors · 2024-11-25
> **Responses for Q5**
>
> Q5: Which specific challenges it addresses and the unique values.
>
> A5: Our benchmark is indispensable because it goes beyond simply combining existing datasets, offering unique contributions that address critical gaps in tool-use evaluation. Unlike previous datasets that primarily focus on synthetic dialogues or idealized scenarios, our benchmark is grounded in real-world user dialogues, ensuring alignment with practical tool-use demands. By synthesizing APIs from real-world data rather than generating dialogues from APIs, we ensure better representation of real-world challenges, as validated by our experimental results in A4.
> While some previous datasets cover similar scenarios, our benchmark emphasizes comprehensive evaluation across single-turn, multi-turn, single-tool, and multi-tool cases, with hard and OOD scenarios designed to test models under realistic complexities. We further provide granular error analyses, including format, action, and parameter errors, offering actionable insights for improvement.
> These unique features ensure that our benchmark not only highlights the limitations of current LLMs but also provides a roadmap for advancing tool-use capabilities, making it an essential resource for the research community.
>
> Thanks again for your kind review.

---

> ### Author Response · Authors · 2024-11-28
>
> Dear Reviewer VABn,
>
> Thanks for your advice. We believe we have addressed your concerns carefully. If you have other questions or comments, please let us know.

---

> > ### Author Response · Authors · 2024-12-02
> >
> > Hi, Reviewer VABn,
> >
> > Thanks again for your insightful and constructive suggestions. As the discussion deadline is coming, please let us know whether our responses have addressed your concerns.
> >
> > If you have other questions, we are glad to give quick feedback.

---

> > > ### Author Response · Authors · 2024-12-03
> > >
> > > Dear **Reviewer VABn**,
> > >
> > > As the discussion deadline is coming, please let us know whether our responses have addressed all your concerns. Besides, if you think that we have solved your questions well, could you reevaluate our work and change your rating?
> > >
> > > Moreover, we believe that our submitted paper has improved a lot based on your insightful and constructive comments.
> > >
> > > Thanks again for your valuable efforts.

---

> > > > ### Author Response · Authors · 2024-12-03
> > > >
> > > > Dear Reviewer VABn,
> > > >
> > > >
> > > > Thank you for your valuable comments. As the discussion period is coming to a close, we would appreciate it if you could let us know whether our responses have addressed your concerns.

---

### Official Review · Reviewer_gpvW · 2024-11-02

**Soundness:** 3
**Presentation:** 3
**Contribution:** 3
**Rating:** 6
**Confidence:** 4

**Summary:**

This paper presents MTU-Bench, a benchmark designed to evaluate large language models (LLMs) in tool-use across diverse and complex dialogue settings, including single-turn, multi-turn, single-tool, and multi-tool tasks. MTU-Bench addresses limitations in existing benchmarks by incorporating automated, cost-effective metrics that do not require GPT-based evaluations. Key contributions include a large dataset of tool-use dialogues synthesized and validated with GPT-4, a detailed evaluation framework (MTU-Eval) with fine-grained metrics, and the introduction of MTU-LLaMA, a model fine-tuned for tool-use tasks that shows strong performance. This work provides a comprehensive benchmark that captures real-world complexities, supporting future advancements in tool-using LLMs.

**Strengths:**

MTU-Bench introduces a unique multi-granularity benchmark for tool-use in LLMs, covering single/multi-turn and single/multi-tool tasks, addressing limitations of prior benchmarks with a cost-effective, automated evaluation approach. It includes detailed metrics (e.g., tool and parameter selection accuracy, task process rate) that provide deep insights into model performance across diverse scenarios, adding rigor to the evaluation process.

**Weaknesses:**

1.The paper’s result analysis could be more comprehensive. While it presents performance comparisons across different models and scenarios, it lacks deeper exploration into the causes behind inconsistent model results, particularly in challenging multi-tool and multi-turn settings. A more granular investigation into factors such as specific error patterns, model architecture differences, or the influence of training data could provide actionable insights. This would help identify underlying reasons for performance variability and guide targeted improvements in model design and training strategies.

2.The paper does not indicate whether the experiments were conducted multiple times or if statistical confidence measures were applied to the results. Without multiple runs or confidence intervals, the stability and reliability of the reported results are uncertain, particularly in complex, multi-turn, multi-tool scenarios where model performance can vary significantly. This omission limits the ability to assess whether observed differences between models (e.g., GPT-4 vs. MTU-LLaMA) are statistically significant or simply due to random variation. Conducting multiple experimental runs and reporting average results with confidence intervals would strengthen the reliability of findings and clarify performance comparisons across different settings.

**Questions:**

1.You classified the types of model errors (such as operation errors, parameter errors, format errors), but did not delve into the specific reasons why the errors occurred or possible resolution strategies. Can the model's specific error patterns under different task complexity or tool combinations be further analyzed to help improve the model?

2.As a binary indicator, the success rate requires that the entire conversation is completely error-free to be successful, which may be too stringent for multi-round conversations. In practical applications, some small errors may not significantly affect the completion of the final task, especially in scenarios where users can tolerate partial mistakes. Such strict standards will cause the model to be marked as a failure due to small errors even at high performance, failing to reflect the overall effect of the model. Therefore, is it necessary to introduce a fault-tolerant mechanism for SR, such as allowing a small number of insignificant errors to exist?

3.SATS uses an exponential decay method to reduce the impact of early errors on subsequent rounds. While this decay mechanism captures the temporal location of errors, it may not be effective enough to cope with the impact of different types of errors. For example, some errors (such as parameter errors) may invalidate the entire conversation, while others (such as minor tool selection errors) have less impact on subsequent conversations. Is it possible to incorporate error type and severity into the decay function to get a more precise round success rate?

4.Current multi-round evaluation metrics do not differentiate between the type of error (e.g. tool selection error, parameter selection error, etc.) and severity. However, different error types have significantly different effects on dialogue. For example, parameter errors often have a greater impact than tool selection errors because parameter errors can lead to complete failure of the task. Therefore, should error type and severity be included in the assessment and given different weights, thereby improving the accuracy of the assessment?
5.In order to ensure the validity of the results in real applications, are there any plans to introduce a part of real human-labeled data and compare the performance difference of the model on real data and synthetic data?

---

> ### Author Response · Authors · 2024-11-25
> **Responses for Q1-Q3**
>
> Thanks for your kind feedback. Here are some explanations that we hope could address your concerns.
>
> Q1: More comprehensive analysis.
>
> A1: We highlight the current limitations of LLM tool-use in Figure 7 and Tables 13, 14, 15, 16. The error pattern can be simply categorized into three error types: (1) format error, which relies on the models’ strong instruction-following capabilities; (2) action error and (3) parameter error which relies on the models’ abilities for code reasoning and long-context understanding abilities especially in multi-turn multi-tool settings. These error patterns can be further broken down into tool selection errors, parameter omissions, parameter misalignments, repeated tool calls, parameter hallucinations, etc. (Tables 13, 14, 15). Moreover, the performance gap between ToolLLaMA and MTU-LLaMA demonstrates that synthesizing APIs from real-world dialogues is more effective than synthesizing dialogues from real-world APIs for real-world tool-use user cases. These findings inspire the future work to improve their data synthesis workflow for tool-use training, and to pay more attention to the coding and long-context understanding abilities of LLMs.
>
> Q2: Statistical confidence and the influence of randomness.
>
> A2: We understand your concern about the confidence in the results. In fact, at the beginning of the experiments, with a rigorous scientific attitude, we establish the principle of conducting each experiment three times. We regret not mentioning this in the paper, but the variance values observed from the three repetitions of each experiment were all within a reasonable range. It indicates that the results are not random variations but are statistically significant. Below, we present part of our experimental records, including the variance results of the average scores of several models across four settings of the normal-set.
>
> |  | S-S | S-M | M-S | M-M |
> | --- | --- | --- | --- | --- |
> | GPT-4 | 1.16 | 1.98 | 1.68 | 1.91 |
> | GPT-3.5 | 2.67 | 1.01 | 0.12 | 0.80 |
> | Qwen2-7B-Instruct | 1.18 | 0.30 | 0.53 | 0.58 |
> | LLaMA-8B-Instruct | 0.21 | 0.03 | 0.05 | 0.03 |
> | MTU-LLaMA | 0.04 | 0.03 | 0.05 | 0.03 |
>
> Q3: Reasons for the errors and the possible solutions.
>
> A3: We agree that a deeper analysis of error causation and potential resolution strategies can significantly enhance the practical utility of our work. While we broadly categorize error patterns into format, action, and parameter errors in the main text, a detailed breakdown is provided in Appendix E. We have already provided a detailed breakdown of error patterns across task complexity and tool combinations, as outlined in Table 8, 13, 14, 15, 16, and Figure 7. Below, we elaborate on how these analyses connect to error causation and resolution strategies.
>
> 1. Error Causation:
>     - Single-Turn Single-Tool Errors: These errors primarily arise from tool selection errors, parameter omissions, and parameter misalignments (Table 13).
>     - Multi-Turn Single-Tool Errors: Increased task complexity introduces issues like repeated tool calls, parameter inheritance failures and hallucinations due to incomplete memory integration over the dialogue context.
>     - Single-Turn Multi-Tool Errors: The calling of fewer tools, more tools, or wrong tools.
>     - Multi-Turn Multi-Tool Errors: The highest complexity, these scenarios combine challenges from all above settings, compounded by the need for robust long-context reasoning and decision-making over interdependent tool usages.
> 2. Possible Solutions:
>     - Improving Instruction-Following for Formatting.
>     - Enhancing Long-Context Understanding.
>     - Reduce tool hallucinations by  incorporating advanced retrievers or leveraging better retrieval-augmented generation techniques.
>     - For parameter errors stemming from omissions, hallucinations, ambiguities, or complex reasoning, we encourage the community to focus on enhancing models' multi-hop reasoning and retrieval capabilities
>
> We believe these insights, coupled with the detailed error categorization and our proposed strategies, can help inform future research in LLM tool-use. This breakdown not only highlights the current limitations but also suggests promising directions for improving real-world model utility.

---

> ### Author Response · Authors · 2024-11-25
> **Responses for Q4-Q7**
>
> Q4: Fault-tolerant mechanism for SR.
>
> A4: The binary Success Rate (SR) is indeed stringent, as it requires a completely error-free conversation to count as successful. To address the concern of fault tolerance in multi-turn conversations, we have introduced complementary metrics such as **Averaged Turn Success Rate (ATS)** and **Soft Averaged Turn Success Rate (SATS)**.
>
> - **ATS** averages the success of each turn, providing a more nuanced measure that gives partial credit for successful turns even if the entire session is not flawless.
> - **SATS** builds on ATS by penalizing errors based on their position in the dialogue, reflecting the intuition that earlier mistakes disrupt task completion more significantly than later ones. This mechanism introduces a level of fault tolerance, accommodating minor or late-stage errors that do not severely impact the final task outcome.
>
> These metrics collectively offer a more comprehensive evaluation framework, balancing the strictness of SR with the practical considerations of fault tolerance in real-world applications. We believe that SATS, in particular, directly addresses your concern by allowing small, insignificant errors to exist while still reflecting the overall effectiveness of the model.
>
> Q5 & Q6: Decay function and type weighting for multi-turn.
>
> A5 & A6: We appreciate your suggestion to incorporate error types and severity into the decay function for a more nuanced multi-turn evaluation. While our current metrics do not explicitly differentiate between error types in weighting or decay, we have analyzed the distribution and impact of different error types in Table 16. For instance, parameter errors occur more frequently than tool selection errors (action errors) in single-turn settings, whereas the opposite is observed in multi-turn settings. This highlights the varying influence of error types depending on dialogue complexity.
>
> To further explore error-specific impacts, we categorized tool selection errors into two groups: (1) **Operative tools**, where errors can cause significant real-world consequences (e.g., incorrect deletions or updates); and (2) **Informational tools**, where errors primarily disrupt subsequent turns by failing to provide critical information. In GPT-4’s results for multi-tool single-turn (M-S) settings, the ratio of informational tool selection errors to operative tool selection errors is 67.65% vs. 32.35%. This indicates that informational errors dominate in these scenarios and can significantly disrupt parameter generation or future calls.
>
> While we have not yet incorporated specific weights or decay adjustments for error types, our analysis of error frequencies and their implications offers valuable insights. Incorporating such mechanisms in future iterations could provide a more precise and context-sensitive evaluation, aligning closely with the impact of different error types on overall task success.
>
> Q7: Introduce some real human-labeled data.
>
> A7: Our dataset is inherently based on real human-labeled data, ensuring its alignment with real-world application scenarios. The dialogues in our benchmark are sourced from real-world human interactions in widely recognized datasets, such as MultiWOZ[1] and SGD[2], which naturally reflect real user behavior and realistic communication patterns.
>
> To adapt these dialogues for tool-use evaluation, we restructured them using GPT models. However, this process focused solely on formatting the dialogue for tool using while keeping the original semantics and intent unchanged (see more from the A4 for Reviewer VABn). To ensure the quality of the data, we also conducted rigorous human quality reviews.
>
> [1] MultiWOZ: a large-scale multi-domain wizard-of-oz dataset for task-oriented dialogue modelling
>
> [2]SGD: Towards scalable multi-domain conversational agents: The schema-guided dialogue dataset
>
> Thanks again for your valuable suggestions.

---

> > ### Comment · Reviewer_gpvW · 2024-11-26
> >
> > Thank you for your response. I appreciate the effort you have taken to address the concerns raised, particularly regarding statistical confidence, error analysis, and the use of complementary metrics. I will maintain my original score.

---

### Official Review · Reviewer_JTwk · 2024-11-04

**Soundness:** 2
**Presentation:** 2
**Contribution:** 2
**Rating:** 6
**Confidence:** 4

**Summary:**

This paper proposes a multi-granularity tool-use benchmark for large language models, called MTU-Bench.

The main contribution of this paper can be summarized as,

-  a novel automated data synthesis pipeline is designed to generate high-quality, fine-grained tool-use datasets from pre-existing task-oriented dialogue datasets.

- introduce MTU-Instruct and MTU-Eval.

Comprehensive experimental results demonstrate the effectiveness of the proposed MTU-Bench.

**Strengths:**

The strengths of this paper are summarized as follows,

- The proposed MTU-BENCH cover more tool-use scenes than previous datasets.

- The evaluation cost of MTU-BENCH is cheaper.

- The authors will open-source all relevant code and data, supporting further research in this field by the community.

**Weaknesses:**

The questions, concerns and weaknesses of this paper are summarized as follows,

- In Section 2.1.1, during the Data Collection phase, the authors should provide a more detailed and comprehensive list of the specific criteria and standards used for dataset selection.

- There appear to be situations where multiple classification criteria, such as 'Information missing,' 'Information exists,' 'Information confirmed,' 'Aimless chatting,' and 'Specific API call,' could apply simultaneously. How should these cases be handled?

- Any visualization results with specific examples can be shown for Tool Clustering in Section 2.1.1?

- Is data quality truly assured? Since the data is synthesized by GPT-4 and also validated by GPT-4, can the reliability of the synthetic data be guaranteed?

- The overall presentation of Section 2.1.1 is not very strong, and many details are not clearly explained (such as quality filters and adjustments). The authors should refine this section thoroughly.

- The content in Section 2.1.2 does not fully align with what is presented in the introduction. The authors should add a reasonable comparison with previous datasets at an appropriate place in the paper.

- Could the authors provide some experimental results that train other open-sourced LLM on MTU-BENCH?

**Questions:**

I have included the main questions in weaknesses box and the authors can response according to the comments there.

---

> ### Author Response · Authors · 2024-11-25
> **Responses for Q1-Q3**
>
> We're deeply grateful for your thorough review and insightful suggestions. We're grateful for your acknowledgment of the idea and the performance of our work. We appreciate the opportunity to clarify certain issues about our work and have the opportunity to discuss some views on the future of this direction.
>
> Q1: Dataset selection specifics.
>
> A1: To ensure the quality of the data, we established a set of detailed and comprehensive standards during the data collection process. Thank you for your reminder. We have added more detailed criteria for data selection in **Appendix B.** Hope readers can understand the data collection process more clearly. The following are the specific criteria we used when selecting the dataset:
>
> 1. **Exclusion of Unsuitable Intents**. We filter out intents that are not suitable for tool calls, particularly those that are difficult to tackle with external tools. For example, conversations seeking naming suggestions are excluded, as they are inherently challenging to define for tool usage. Manual verification is employed to achieve this, resulting in a 26% data exclusion rate.
> 2. **Redundancy Elimination through Clustering**. The synthesis of tools can lead to redundancies, such as *"share_location", "share_my_location" and "share_current_location"*. To mitigate this, we adopt a clustering approach to group similar tools, retaining only one representative tool from each cluster. We establish a lower threshold for initial clustering to ensure each cluster contains a unique tool, followed by manual elimination of any duplicates. This process achieves a reduction ratio of approximately 20:1, as illustrated in Figure 8.
> 3. **Filtering of Undefined Tools**. About 6% of the synthesized data includes tools that are not defined in the tool library. This data is filtered out using rule-based matching methods.
> 4. **Parameter Correctness Check**. Approximately 16.9% of the synthesized data fails our correctness checks, comprising 3.2% of cases with fabricated non-existent parameters and 13.7% where generated parameters do not meet format validation requirements.
> 5. **LLM Verification**. We utilize an LLM, specifically GPT-4, to recheck the correctness of all answers,  including user demand rationality, tool completeness, observation rationality, parameter validity, response rationality, factual correctness, and semantic coherence. Approximately 10% of data is filtered out during this process. This process results in an additional 10% of data being filtered out.
> 6. **Manual Quality Annotation.** We conduct manual quality checks utilizing multiple experts. For the training set, 500 samples are randomly selected and each is evaluated by three experts, with any discrepancies resolved by a fourth. This approach yields a 96\% accuracy rate. For the test set, all samples are meticulously calibrated by hired experts, achieving a final accuracy of 100%.
>
> Q2: The handling of multiple classification criteria.
>
> A2: Actually, these classification criteria can't apply simultaneously.  We are willing to provide a more detailed explanation of our classification standards. These standards were used when creating tools with GPT-4. It is important to note that we provided GPT-4 with user questions as well as the Assistant's Golden Response (sourced from the original dialogue dataset) for tool creation. However, not every round requires create tools; for example, the current round is Aimless chatting or Information missing. Therefore, we designed the classification rules  to assist with annotation. **The classification is based on the Golden Responses of the Assistant from the original dataset. Each response corresponds to only one specific situation**. As a result, these situations will not appear simultaneously in the classification. We hope our response can clarify your confusion.
>
> Q3: Visualization of tool clustering.
>
> A3: Displaying some clustering visualization results will help readers better understand the process. Based on your recommendation, we create a figure to show the clustering results for several categories. As described in our paper, we used Phrase-BERT to extract embeddings for the names of tools and applied a fixed distance threshold for clustering. To facilitate visualization, we use PCA to reduce the dimensionality of the embeddings. We present the clustering results for partial samples from five categories: adjust_lighting, add_new_contact, set_music_volume, share_location, and get_sport_info. Unfortunately, we are unable to upload the image here. We have included these visualization results in the **Figure 8 of Appendix B**, and you can view them in the latest revision.

---

> ### Author Response · Authors · 2024-11-25
> **Responses for Q4-Q7**
>
> Q4: Data quality and reliability.
>
> A4: Data quality is very crucial for us, so we implement comprehensive quality control processes during the data collection phase as follows:
>
> 1. We have established a detailed set of **standards to ensure the quality of the selected dataset**. You can find the specific details in the response to the first question.
> 2. We also verify the quality of data through **manual annotation**. Specifically, we have hired multiple experts to conduct manual quality checks based on similar principles. For trainset, we random selected 500 samples, and then each sample was checked by three experts, and any discrepancies in labeling were resolved by a fourth expert. Finally, we achieved 96% accuracy in the 500 training samples. For the testset, we also hired experts to calibrate the testset samples to ensure that all samples are correct. After manual verification, the accuracy of the test set is 100%.
> 3. We acknowledge that ensuring the complete accuracy of all data is very difficult. In the experiment section, we also observe **significant performance improvements** are obtained after training on our training set, which also indicates that our training set is sufficient to improve the tool-use abilities of LLMs.
>
> Q5: Refinement of Section 2.1.1.
>
> A5: We have revised Section 2.1.1, specifically supplementing the processes for quality filtering and adjustments in Appendix B. You are able to view these details in the latest version.
>
> Q6: Alignment between Section 2.1.2 and the introduction.
>
> A6: We appreciate your comments regarding the alignment between Section 2.1.2 and the introduction. In fact, both our Introduction and Section 2.1.2 aim to demonstrate the superiority of our data in terms of the number of dialogues, various settings, and RealWorld data sources. In Section 2.1.2, we provide some specific metrics across different dimensions, such as Figure 4, which shows the scale of the dataset. Due to space limitations, we have placed some information in Appendix C. For example, Table 7 displays our multi-setting configurations. We further improved the clarity of our description in Section 2.1.2.
>
> Q7: Other backbones.
>
> A7: In addition to the experimental results of LLaMA3-8B presented in our paper, we also provided results for the LLaMA2 series models, including LLaMA2-7B, LLaMA2-13B, and LLaMA2-70B. Please refer to Figure 6 for these results.
>
> We have further supplement the experimental results of LLaMA3-70B and the recently released Qwen2.5 (including Qwen2.5-7B and Qwen2.5-72B) after training on MTU-Instruct. Some of the results are shown below:
>
> Table : Average Scores of the models in normal set of four settings.
>
> |  | S-S | M-S | S-M | M-M |
> | --- | --- | --- | --- | --- |
> | Qwen2.5-7B-Instruct | 61.61 | 41.35 | 30.17 | 24.74 |
> | MTU-Qwen2.5-7B | **70.19(+8.58)** | **51.90 (+10.55)** | **33.00 (+2.83)** | **37.07 (+12.33)** |
> | Qwen2.5-72B-Instruct | 74.11 | 51.55 | 55.55 | 50.87 |
> | MTU-Qwen2.5-72B | **75.96 (+1.85)** | **61.68 (+10.13)** | **59.56(+4.01)** | **56.07 (+5.20)** |
> | LLaMA3-70B-Instruct | 72.12 | 50.88 | 29.76 | 22.99 |
> | MTU-LLaMA3-70B | **72.27 (+0.15)** | **56.64(+5.76)** | **58.08 (+28.31)** | **42.53 (+19.56)** |
>
> The results show that all models exhibit significant improvements over their base models across various settings. We have uploaded detailed experimental results in **Appendix E** of the latest version of the paper. Thank you again for your question.

---

> ### Author Response · Authors · 2024-11-28
>
> Hi, Reviewer JTwk,
>
> We believe we have addressed your concerns carefully. If you have other questions or comments, please let us know. We are very glad to solve your concerns.
>
> Thanks for your insightful suggestions.

---

> > ### Comment · Reviewer_JTwk · 2024-11-30
> >
> > Thanks for the clarification. Considering the content of the authors' rebuttal, I would like to increase the score accordingly

---

### Author Response · Authors · 2024-11-25
**Summarization of the Responses**

Thanks for handling/reviewing our submitted manuscript: "**MTU-Bench: A Multi-granularity Tool-Use Benchmark for Large Language Models**". We would like to thank the reviewers for their insightful and constructive comments and suggestions. By addressing each of the issues raised by the reviewers, we believe that the quality and clarity of our MTU-Bench can be improved a lot. The major responses are summarized as follows:

(1) We have supplemented the detailed process of our data selection, which ensures the quality of our final data. (See Reviewer JTwk.Q1&Q4, Reviewer p9mu.Q6)

(2)We have added more performance comparisons of open-source models before and after training. (See Reviewer JTwk.Q7)

(3) We have provided more clarification and discussion on the motivation and completeness of our metrics. (See Reviewer gpvW.Q4, Reviewer gpvW.Q5&Q6, Reviewer p9mu.Q3)

(4) We have clarified the characteristics of our data originating from real-world sources.(See Reviewer gpvW.Q7, Reviewer VABn.Q4) We have compared our tools and real-world tools (See Reviewer VABn.Q4), validating the effectiveness of our metrics for evaluating real tools.(See Reviewer p9mu.Q5)

(5) We have provided further clarification on the motivation, novelty, and contributions of our research. (See Reviewer VABn.Q1&Q5)

(6) We have discussed more the underlying reasons behind the model's performance and the directions for future improvement. (See Reviewer VABn.Q2&Q3)

(7) We have discussed the model's ability to recover from errors on MTU-Eval. (See Reviewer p9mu. Q2)

(8) We have significantly improved the writing quality based on the writing suggestions and chart presentations in the comments. (See Reviewer JTwk.Q3&Q5&Q6).

---

### Author Response · Authors · 2024-11-25
**Looking forward to feedback on the Responses.**

Dear Reviewers:

Hello! We have updated the responses to your constructive and insightful comments, and we would like to kindly ask you to take a look at our responses and reevaluate our work based on our clarifications. Please let us know whether our response addresses your concerns or whether there is any further detail we can provide to help address them. We appreciate your time and consideration!

---

### Author Response · Authors · 2024-11-27

Dear Reviewers:

Hello! As the discussion phase is short, we would like to ask you to review our responses and reevaluate our work based on our clarifications. We are looking forward to your insightful feedback if you have additional comments. Again, we appreciate your valuable time and consideration!

---

### Comment · Area_Chair_YJeV · 2024-11-29

Dear authors and reviewers,

This is a reminder that the interactive discussion period will conclude in three days (December 2nd). If you haven’t already, please take this opportunity to review the responses and provide any feedback or clarifications.

Thank you,
AC

---

### Meta-Review · Area_Chair_YJeV · 2024-12-23

**Metareview:**

The paper introduced a new benchmark MTU-Bench for evaluating the LLM's abilities to use tools in multiple scenarios. MTU-Bench considers more granular settings in aspects of the number of tools that can use and the number of rounds of tools can use in multi-turn conversation. The authors also provide a pipeline to construct such usage scenarios from existing high-quality datasets.
They also present MTU-Eval for automated evaluation and MTU-Instruct for tool use fine-tuning.

The proposed multi-granularity tool-use benchmark is new and useful. It provides a practical, i.e., low cost, and comprehensive evaluation of existing tool usage capabilities. The paper provides a wide evaluation and analysis of multiple LLms across scenarios.

The rebuttal has addressed most of the concerns and was acknowledged by several reviewers. The only reviewer who gave a rating of 5 didn't respond and I believe most of the concerns can be addressed by the rebuttal and other reviews.

I recommend accepting this paper. This work can help push the development and exploration of tool-use capabilities of LLMs, which is important for future LLM applications.

**Additional Comments On Reviewer Discussion:**

The initial ratings from the reviewers are 5, 6, 5, 5.

The main concerns include:
- Reviewer JTwk: dataset selection specifics, handling of multiple classification criteria, data quality and reliability, writing, experiments on other open LLMs.
- Reviewer gpvW: insufficient comprehensive analysis, statistical confidence and the influence of randomness, fault-tolerant mechanism.
- Reviewer VABn: limited novelty, lack of real-world impact, insufficient analysis, real-world applicability.
- Reviewer p9mu: unable to cover complex tool-calling scenarios,  insufficient designed evaluation metrics, not real-world APIs, the validity of tools, API docs fall into the GPT-4 distribution, limited discussion of current tool-use benchmarks.

The authors provided additional experiments to validate the soundness of the proposed benchmark, and more analysis for more comprehensive evaluation.
The rebuttal is well accepted by the reviewers.
After the rebuttal, the final ratings are 6, 6, 5, 6.
JTwk, p9mu changed the ratings from 5 to 6. gpvW maintained the score of 6.  VABn with rating 5 didn't respond. After reading the rebuttal and reviews, I think the authors has addressed the points raised by VABn.

---

### Decision · Program_Chairs · 2025-01-22

Accept (Poster)